# Leveraging spatial transcriptomics data to recover cell locations in single-cell RNA-seq with CeLEry

Qihuang Zhang [1,6] ✉, Shunzhou Jiang[2,6], Amelia Schroeder[2,6], Jian Hu [3], Kejie Li[4], Baohong Zhang[4], David Dai[5], Edward B. Lee [5], Rui Xiao [2] & Mingyao Li [2] ✉

Single-cell RNA sequencing (scRNA-seq) has revolutionized our understanding of cellular heterogeneity in health and disease. However, the lack of physical relationships among dissociated cells has limited its applications. To address this issue, we present CeLEry (Cell Location recovEry), a supervised deep learning algorithm that leverages gene expression and spatial location relationships learned from spatial transcriptomics to recover the spatial origins of cells in scRNA-seq. CeLEry has an optional data augmentation procedure via a variational autoencoder, which improves the method's robustness and allows it to overcome noise in scRNA-seq data. We show that CeLEry can infer the spatial origins of cells in scRNA-seq at multiple levels, including 2D location and spatial domain of a cell, while also providing uncertainty estimates for the recovered locations. Our comprehensive benchmarking evaluations on multiple datasets generated from brain and cancer tissues using Visium, MERSCOPE, MERFISH, and Xenium demonstrate that CeLEry can reliably recover the spatial location information for cells using scRNA-seq data.

Knowledge of the spatial origins of cells is essential for understanding the spatial organization of tissues and communications among cells. While single-cell RNA sequencing (scRNA-seq) has made it possible to characterize cell types and states at an unprecedented resolution, the lack of physical relationships among cells has limited its applications. Recently developed spatial transcriptomics (ST) technologies can overcome the limitations of scRNA-seq by profiling transcriptome-wide gene expression while retaining the location information of measured genes within tissues[1–7]. ST provides complementary information to scRNA-seq of dissociated cells regarding the relationships between gene expression and spatial locations. Since a large amount of scRNA-seq data have been generated, it is desirable to recover their cell location information by leveraging gene expression-spatial location relationships learned from ST. The location recovered scRNA-seq data can be utilized in downstream analysis, e.g., cell-cell communications, in which knowledge of cell locations is important.

The earliest work for spatial location recovery of scRNA-seq data dated back to Seurat[8], which recovers the locations of cells by modeling information provided by anchor genes with known locations. While Seurat has shown promising performance for scRNA-seq data generated from zebrafish embryos, the reliance on location-specific anchor genes has limited its applications since anchor gene information is often unavailable. On the other hand, ST platforms, such as 10x Visium, SLIDE-seq[9], and Stereo-seq[10], provide information about the

[1]Department of Epidemiology, Biostatistics and Occupational Health, School of Population and Global Health, McGill University, Montreal, QC, Canada. [2]Statistical Center for Single-Cell and Spatial Genomics, Department of Biostatistics, Epidemiology, and Informatics, Perelman School of Medicine, University of Pennsylvania, Philadelphia, PA 19104, USA. [3]Department of Human Genetics, School of Medicine, Emory University, Atlanta, GA 30322, USA. [4]Research Department, Biogen, Inc., 225 Binney St., Cambridge, MA 02142, USA. [5]Translational Neuropathology Research Laboratory, Department of Pathology and Laboratory Medicine, University of Pennsylvania, Philadelphia, PA 19104, USA. [6]These authors contributed equally: Qihuang Zhang, Shunzhou Jiang, Amelia Schroeder. ✉e-mail: qihuang.zhang@mcgill.ca; mingyao@pennmedicine.upenn.edu

relationships between transcriptome-wide gene expression and spatial locations. While anchor genes can potentially be identified from ST data, methods that are specifically designed for anchor gene detection are still limited. Most spatially variable gene detection methods aim to detect genes that show spatial variation but do not examine whether the detected genes are location-specific[11–14].

To utilize information in ST, recent methods such as Tangram[15] and CytoSpace[16] have been developed. Although these methods can be utilized to infer the location information for cells in scRNA-seq, cell location recovery is not their primary objective, and hence when applied to the task of cell location recovery, their performance is suboptimal. For example, both Tangram and CytoSpace aim to enhance gene expression resolution and throughput in ST based on spatially mapped single-cell profiles. As such, Tangram assumes the spatially mapped scRNA-seq cell density is similar to that in the ST reference, and CytoSpace assumes that there is a one-to-one mapping between the scRNA-seq data and the ST reference; both assumptions lead to the filtering of many scRNA-seq cells. While the cell density assumption or the one-to-one mapping assumption might be appropriate when the goal is to enhance the ST data quality, it is problematic when the goal is to recover the locations of cells in scRNA-seq. Other methods, such as spaOTsc[17], aim at inferring cell-cell communications from scRNA-seq data. It employs an optimal transport algorithm to construct the spatial mapping of cells. This algorithm is designed for data with small cell numbers and often fails to run when applied to large scRNA-seq datasets. novoSpaRc[18] is another method that utilizes the optimal transform algorithm to spatially map cells to tissue locations. By assuming cells in physical proximity also share similar gene expression profiles, novoSpaRc can recover the spatial locations of scRNA-seq cells without the reliance on a spatial reference, but it can also include a reference atlas of marker genes to improve its performance. Tangram, spaOTsc, and novoSpaRc all map the scRNA-seq cells to multiple locations in a probabilistic fashion. However, these methods treat locations as discrete variables, and the lack of consideration of the physical proximity of nearby locations can make the predicted locations for a given cell to be far apart from each other, which leads to difficulties in interpretation.

To address the limitations of existing methods, we developed CeLEry (Cell Location recovEry), a machine-learning method to recover the location information for cells in scRNA-seq by utilizing the gene expression and spatial location relationships learned from ST. The location information of a cell can be represented at multiple levels. The most precise representation is the 2D location in a tissue section, where the recovery of such information relies on genes whose expression patterns show spatial gradients in the tissue. Another location representation is the spatial domain that a cell resides. Although not as precise as the 2D location, it still provides valuable information when studying human disease. Spatial domain recovery is a much more feasible task than 2D location recovery since such recovery is mainly driven by spatial domain-specific genes, which can be detected using methods such as SpaGCN[14]. CeLEry is able to recover the location information at multiple levels, including the 2D location and the spatial domain of a cell. We also provide uncertainty estimates for the recovered locations. Through comprehensive benchmark evaluations on data generated from brain and cancer tissues, we show that CeLEry can reliably recover the location information of cells and the application to a single-nucleus RNA-seq (snRNA-seq) dataset generated from postmortem brains with Alzheimer's disease (AD) revealed the loss of neuronal cells in specific cortical layers compared to normal brains.

## Results
### Overview of CeLEry
The key idea of CeLEry is to construct a model to characterize the relationships between gene expression and spatial locations based on a training ST dataset, and then apply the fitted model to a scRNA-seq or snRNA-seq test dataset by using gene expression information in the single cells to recover their location information. CeLEry uses a deep neural network to learn the relationships between gene expression and spatial locations by minimizing a loss function that is specified according to the specific problem (Fig. 1a). For 2D location recovery, a mean squared error (MSE) loss is constructed, and an ellipse quantile regression loss function is adopted to estimate the elliptical region of the predicted location. For spatial domain or cortical layer recovery, a logistic loss or a rank-consistent logistic loss function is used. Then, the trained model is applied to predict the spatial location for each cell in the scRNA-seq data (or snRNA-seq data) and quantifies the prediction uncertainty.

The performance of a supervised prediction model depends on the sample size of the training data. Since ST data are still expensive to generate, the sample size of the training data might be limited, which might lead to overfitting. To improve the robustness of the prediction model, CeLEry has an optional data augmentation procedure to generate replicates of the original ST data (Fig. 1b). First, we learn the mean and standard deviation of the gene expression embedding distribution using a variational autoencoder integrated with gene clustering, and then generate a series of gene expression replicates by sampling embeddings from the learned distribution. The data augmentation procedure integrates two extra pieces of information from the original ST data: the spatial relationship among spots and the co-expression relationship among genes, where the former is modeled through a convolutional neural network in the variational autoencoder and the latter is modeled through clustering of genes that show similar spatial gene expression patterns. The gene clustering information is then further integrated into the generating model via the embedding layer. By concatenating the embedding of the variational autoencoder with the gene expression clustering results, we can account for the co-expression relationship among genes.

### Data augmentation generates realistic replicates of ST data
To evaluate the performance of the data augmentation procedure, we analyzed a 10x Visium dataset generated from the mouse posterior brain. Figure 2a shows examples of six randomly selected gene clusters obtained during the gene clustering step, where each row includes four example genes randomly selected from a cluster. To understand the overall gene expression patterns of the clusters, for the above six gene clusters, we visualized the spot-wise mean and standard deviation of gene expression across all genes in each cluster (Supplementary Fig. 1). The standard deviation was rather constant regardless of locations, indicating the genes are well clustered. To illustrate that our data augmentation procedure can generate gene expression replicates that resemble the original ST data, we visualized two randomly selected genes *RPS29* and *CALB1*, in Fig. 2b, c, together with the previously selected representative gene for each of the six gene clusters in Supplementary Fig. 2. These results indicate that the gene expression patterns in the original data and the three replicates generated from our data augmentation procedure are similar. Moreover, the augmentation-generated gene expression shows variations among replicates, which increases the potential of our model to learn more spatial information from the data.

To determine the similarity between the generated replicates and the actual biological replicates, we also analyzed a 10x Visium dataset generated by ref. 19 (denoted as the LIBD data). The LIBD data were generated from three postmortem brains, with each brain having four tissue sections obtained from the dorsolateral prefrontal cortex (DLPFC). This dataset allows us to examine whether the generated replicates from our data augmentation procedure resemble those actual biological replicates. For illustration purposes, we randomly selected three genes *CAMK2N1, TMSB10*, and *HPCA*, and visualized their expression (Supplementary Fig. 3). While the artificial replicates

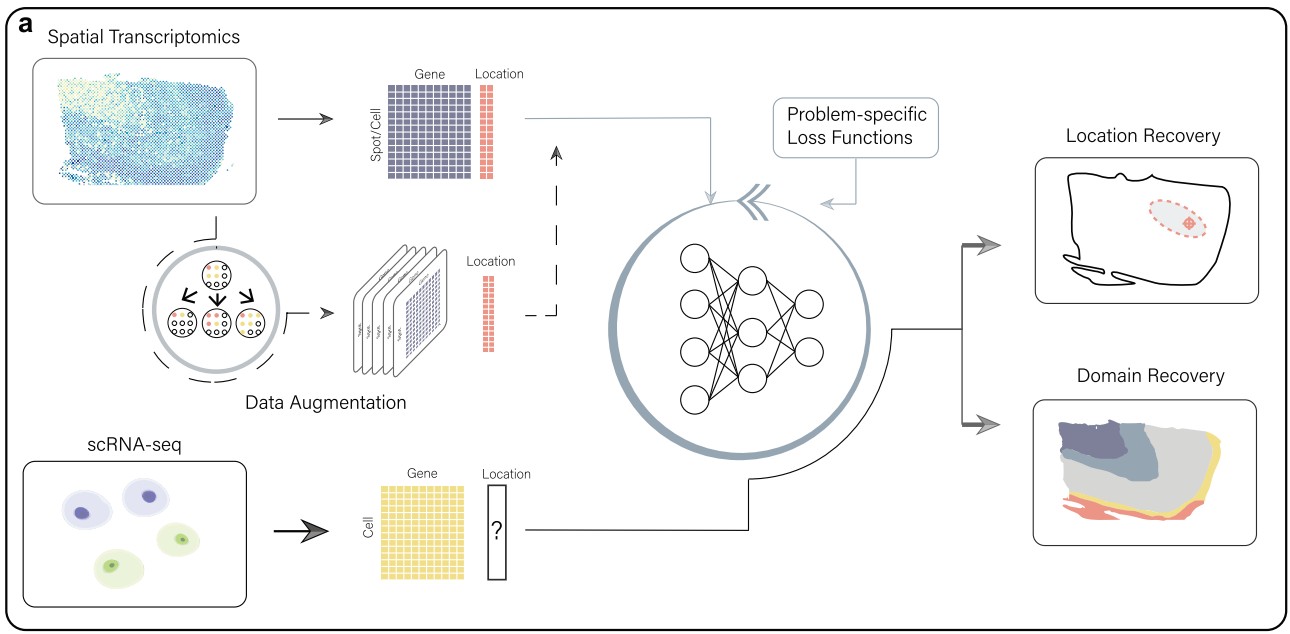

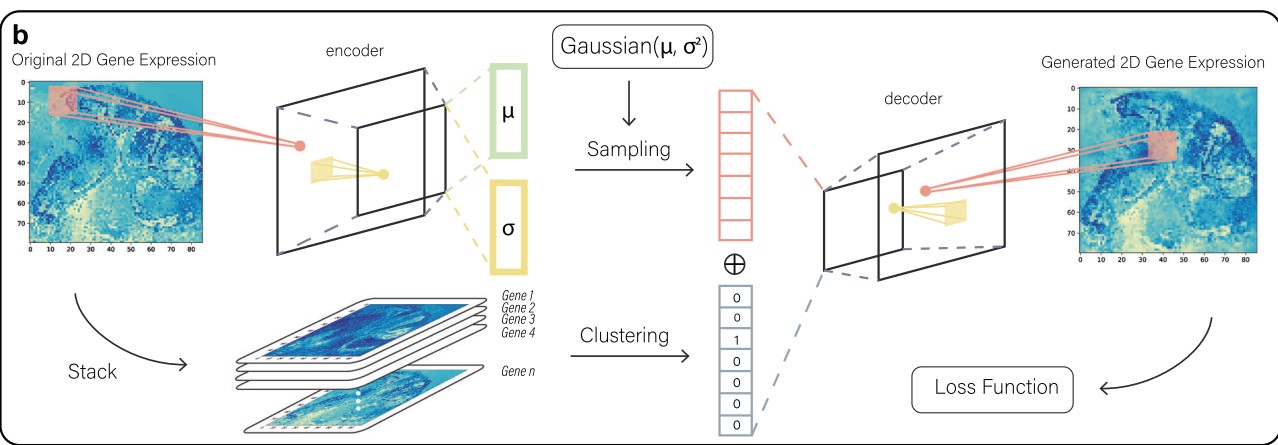

**Fig. 1 | Workflow of CeLEry. a** CeLEry takes an ST dataset as input for model training and a scRNA-seq dataset as input for cell location prediction. CeLEry has an optional data augmentation step, which optionally generates replicates of the ST data via a variational autoencoder. The generated data are then included in the training data. A deep neural network is trained to learn the relationship between the spot-wise gene expression and location information by minimizing a loss function that is specified according to the specific problem. Then, the trained model is applied to the scRNA-seq data to predict the location of each cell. **b** The data augmentation procedure consists of an encoding stage and a decoding stage. In the

encoding stage, the 2D expression pattern for each gene is summarized into the embeddings of mean and standard error vectors of a multivariate normal distribution. In the decoding stage, new embeddings are generated via the multivariate normal distribution given by the encoding stage. Meanwhile, a clustering algorithm is performed to cluster genes into groups. Finally, the generated embedding and the gene cluster embedding are concatenated, which is used as input for a convolutional neural network to decode the concatenated embedding into a 2D matrix with the same dimension as the gene expression input. The resulting 2D matrix can generate replicates of the ST data.

were smoother than the biological replicates, they displayed gene expression variation across replicates. These results indicate that while artificial replicates cannot completely replace biological replicates, they can serve as an alternative for introducing variabilities in the training data when biological replicates are not available. For ST references that have large numbers of spots or cells, data augmentation is not needed.

### Benchmark evaluation for cortical layer recovery in the human dorsolateral prefrontal cortex

To evaluate the performance of CeLEry in recovering spatial domains, we performed benchmark evaluations using the LIBD data. This study has manually annotated cortical layer information, which allows us to evaluate the accuracy of spatial domain recovery. Since there is a natural order among the cortical layers, we considered the order information when recovering the layers in CeLEry by minimizing a

rank-consistent logistic loss function. To evaluate the performance of CeLEry with different degrees of difficulty in cortical layer recovery, we considered four scenarios (Fig. 3a): (1) the training data include one tissue section (sample ID: 151673) that is from the same brain as the test sample (sample ID: 151676); (2) the training data include one tissue section (sample ID: 151673) that is from a different brain as the test sample (sample ID: 151507); (3) the training data include three tissue sections (sample IDs: 151673, 151674, and 151675) that are from the same brain as the test sample (sample ID: 151676); and (4) the training data include three tissue sections (sample IDs: 151673, 151674, and 151675) that are from a different brain as the test sample (sample ID: 151507). These scenarios allow us to evaluate how the performance of CeLEry is impacted by the training sample size and heterogeneity between the training and test data. Of note, novoSpaRc, spaOTsc, and Tangram cannot analyze data in Scenarios 3 and 4 because they can only take one tissue section as the training data.

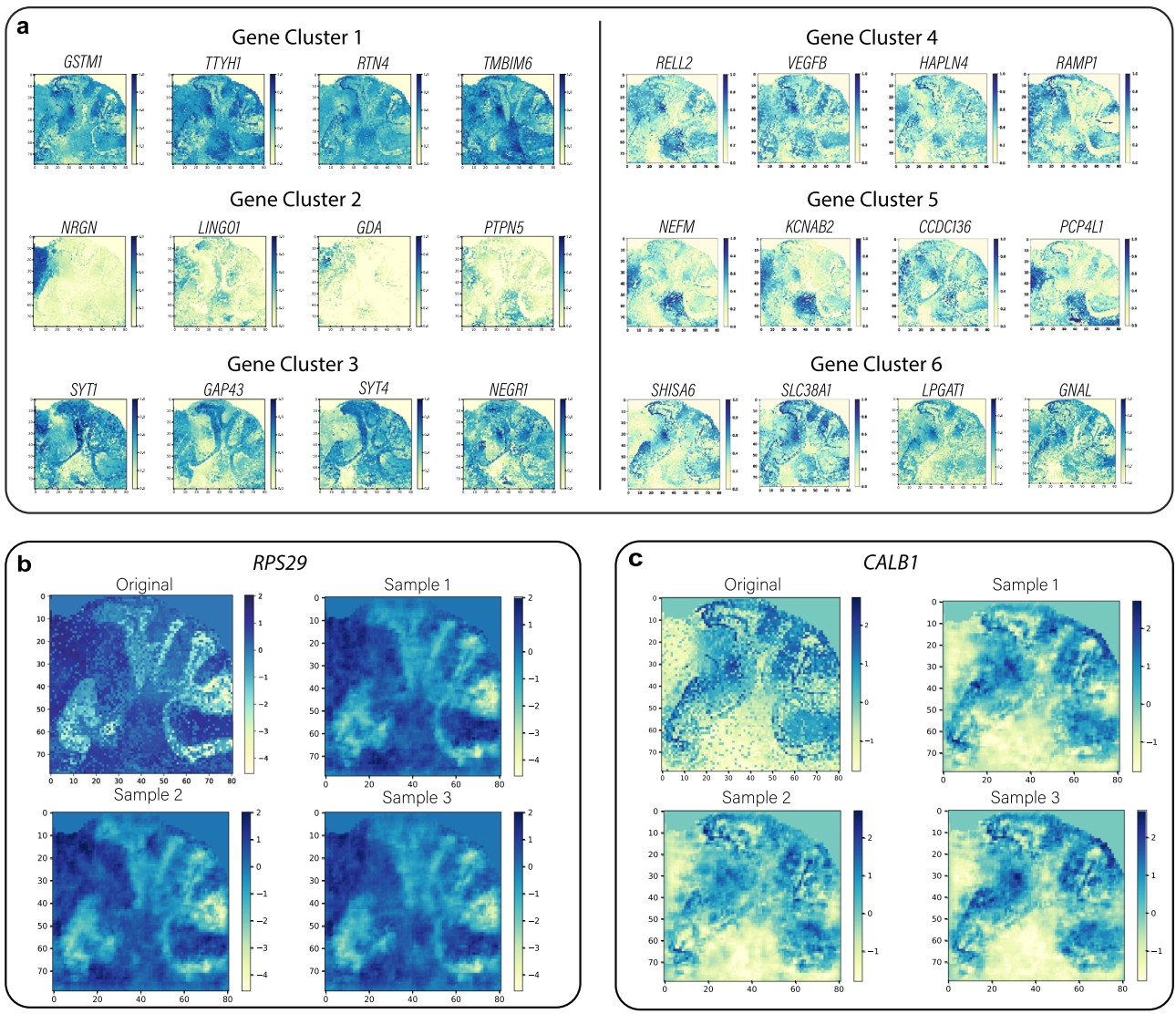

**Fig. 2 | Evaluation of the data augmentation procedure. a** Examples of six gene clusters obtained from the mouse posterior brain ST dataset, where each row contains four representative genes from a cluster. **b**, **c** Examples of 2D gene expression maps for genes *RPS20* and *CALB1*, together with three of their replicates generated from the data augmentation procedure. Source data are provided as a Source Data file.

First, we examined whether the data augmentation step provides useful information in predicting cortical layers when the training data only include one tissue section (Scenarios 1 and 2). Our results show that two replicates are already adequate for improving the prediction model (Fig. 3b). Figure 3c shows the layer-specific prediction accuracy. In general, we found it is relatively easy for CeLEry to recover spots that originate from the white matter, and augmentation helped improve the prediction accuracy for most layers. In contrast, Tangram showed consistently low accuracy for all layers; novoSpaRc predicted with higher accuracy in white matter (WM) but relatively low accuracy in other layers; and spaOTsc, however, predicted most of the spots to L3 or L5 regardless of their true layers, showing imbalanced performance among layers with high accuracies for these two layers only but low accuracies for others.

To assess the overall performance of the prediction (Fig. 3d), we considered both top-1 and top-2 accuracies, where the top-1 accuracy is defined as the percentage of spots that are correctly predicted to the true layer, and the top-2 accuracy is defined as the percentage of spots that are predicted either to the true layer or an adjacent layer. The top-1 and top-2 accuracies for CeLEry were 53.8 and 89.2%, respectively, for Scenario 1, and 44.0 and 82.6%, respectively, for Scenario 2. When the

data augmentation step was included, the top-1 accuracy increased to 63.2% for Scenario 1 and 51.2% for Scenario 2, and the top-2 accuracies increased to 92.1% for Scenario 1 and remained similar for Scenario 2 (81.6%). In general, CeLEry achieved much higher prediction accuracy than Tangram, spaOTsc, and novoSpaRc. For Scenario 1, spaOTsc's top-1 and top-2 accuracies were 49.1 and 71.2%, respectively, novoSpaRc's top-1 and top-2 accuracies were, 44.4 and 71.0%, respectively, and the corresponding accuracies for Tangram are 13.2 and 33.3%. For Scenario 2, Tangram, spaOTsc, and novoSpaRc have even lower accuracies (Tangram: 16.5% for top-1 and 35.3% for top-2; spaOTsc: 39.3% for top-1 and 65.0% for top-2; novoSpaRc: 37.6% for top-1 and 63.8% for top-2).

Next, we investigated the contribution of having additional tissue sections in the training set for prediction (Scenarios 3 and 4). Without data augmentation, the top-1 and top-2 accuracies for CeLEry for Scenario 3 were 67.2 and 96.4%, respectively, and 52.7 and 88.0%, respectively, for Scenario 4. Both Scenarios 3 and 4 have better prediction results than Scenarios 1 and 2, indicating that having additional tissue sections in the training set can improve the prediction accuracy. Interestingly, we found that for Scenario 2, after data augmentation, the top-1 accuracy for CeLEry increased to 51.2%, which is similar to the 52.7% top-1 accuracy for Scenario 4 (Fig. 3d). Similar performance was

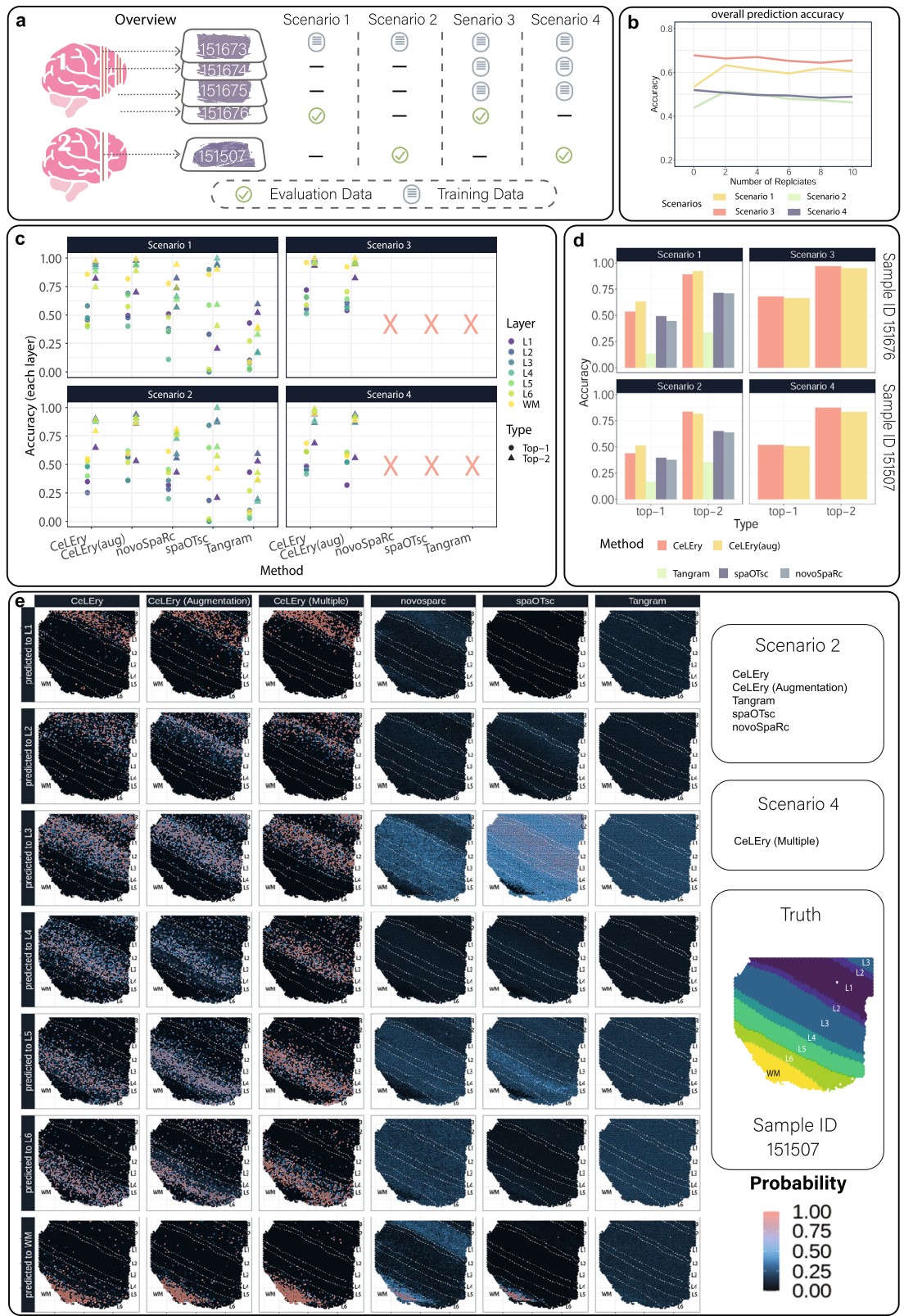

observed between Scenario 1 with data augmentation (63.2% top-1 accuracy) and Scenario 3 (67.2% top-1 accuracy). In particular, the augmented CeLEry had better performance in recovering L2, L3, L5, and L6 than the ordinary CeLEry and showed comparable performance to Scenario 4 when multiple training tissue sections were available (Fig. 3e). These results suggest that data augmentation can serve as a useful alternative when limited training tissue sections are available.

## Application to recover cortical layers for snRNA-seq data in AD brains

To illustrate the recovered cortical layer information is useful for disease-based studies, we analyzed an snRNA-seq dataset generated from an Alzheimer's disease (AD) study by ref. 20, which includes 122,606 cells obtained from the frozen middle frontal neocortex of 15 postmortem brains with varying pathologies, but matched for sex and

**Fig. 3 | Cortical layer recovery for spots in the LIBD human DLPFC data. a** Four scenarios were considered in the evaluation. These scenarios vary in the number of tissue sections in the training data and the source of the test data, representing situations with different degrees of location recovery difficulty. **b** The overall layer prediction accuracy for Scenario 2 with different numbers of replicates obtained from the data augmentation procedure. **c** Layerwise prediction accuracies under different scenarios using CeLEry without data augmentation, CeLEry with data augmentation (two replicates), novoSpaRc, spaOTsc, and Tangram. The results for Tangram are missing for Scenarios 3 and 4 because Tangram can only take one tissue section as the training data. L1: layer 1, L2: layer 2, L3: layer 3, L4: layer 4, L5: layer 5, L6: layer 6, WM white matter. **d** Overall top-1 and top-2 prediction accuracies for CeLEry, CeLEry with data augmentation, Tangram, spaOTsc, and

novoSpaRc, under different scenarios. It is noteworthy that Tangram, spaOTsc, and novoSpaRc are not applicable for Scenarios 3 and 4. **e** Visualization of the probabilities of assigning each spot to different layers (shown as different rows) in using CeLEry without data augmentation (Scenario 2), CeLEry with data augmentation (Scenario 2, 2 replicates), CeLEry with multiple training samples (Scenario 4), novoSpaRc, spaOTsc, and Tagram (Scenario 2). For Tangram, novoSpaRc, and spaOTsc, the probability of predicting a cell to be in a layer is calculated by summing up the probabilities of predicting the cell for all spots belonging to that specific layer. The ground truth cortical layer structure for the test sample is shown on the right. L1–L6 and WM were defined similarly as in Fig. 3c. Source data are provided as a Source Data file.

age. Clustering using DESC[21] and cell type annotation using marker genes inferred seven cell types in this dataset, including microglia (Mic), excitatory neurons (Ex), inhibitory neurons (In), astrocytes (Ast), oligodendrocytes (Oli), oligodendrocyte progenitor cells (Opc), and endothelial cells (Endo). We used four tissue sections obtained from the same brain (sample IDs: 151673, 151674, 151675, and 151676) in the LIBD human DLPFC data[19] as the training set to learn the relationships between gene expression and cortical layers (L1–L6 and WM) and applied the learned model to recover the layers for cells in the snRNA-seq data into L1–L6 or WM (Fig. 4a). Since the DLPFC data and the snRNA-seq data were obtained from different disease conditions, we only considered genes that were not differentially expressed between the control and AD cells in the snRNA-seq dataset when building the CeLEry cortical layer prediction model. The overall layer recovery results were shown in Fig. 4b, in which we counted the number of cells that are mapped to each layer with the maximum mapping probabilities. Interestingly, the uncertainties of cell predictions were different across different layers but less so across different cell types (Fig. 4c). Our analysis identified varying compositions of cell types in each layer (Fig. 4c), with L1 being dominated by astrocytes, whereas L2-L5 had more neuronal cells. The oligodendrocytes were mainly mapped to L6 and especially the white matter.

AD is known to be associated with a notable decrease in neurons[22]. Next, we assessed the spatial distribution of the neurons stratified by pathology defined in ref. 20 (Fig. 4d). AT scores, where the presence of amyloid was designated A+ and tau tangles were designated T+, were assigned to each of the 15 brains by two independent neuropathologists blinded to clinical data. Among the 15 brains, eight were A+T+, five were A+T− group, and two were A−T−. Comparing the A+T− brains with those A−T− brains, we found evidence of neuron depletion in the deeper neocortical layers, such as L5 and L6, which is consistent with the distribution of neurotic plaques in AD. As the disease progresses from A+T− to A+T+, more neurofibrillary tangles (tau) accumulate, causing neuronal cell loss involving not only L5 and L6 but also L4[18]. Similar patterns were found when comparing the A+T+ group with the A−T− group, with the A+T+ brains having a decreasing proportion of neurons in L4, L5, and L6, where L4 and L5 had the strongest implication of neuronal cell loss, which is consistent with tau accumulation in L3 and L5. For the inhibitory neurons, cell loss in L3 was also observed.

### Benchmark evaluation for 2D location recovery in mouse brain with 10x Visium data as spatial reference

Encouraged by the previous cortical layer predictions, we next considered a more challenging problem, where the goal is to recover the 2D location of a cell. The results from such analysis can facilitate the study of cell-cell communications in scRNA-seq. We assessed the performance of CeLEry in recovering the 2D coordinates for spots in the mouse posterior brain data generated by 10x Visium. As 2D location recovery is a much more challenging task than cortical layer and spatial domain recovery, having a training set with relatively higher-resolution gene expression is important. To this end, we first enhanced

the gene expression resolution of the ST training data using TESLA[23]. We selected 358 genes according to the spatially variable genes detected by spaGCN[14] and generated super-resolution gene expression for these genes at the super-pixel level (50 × 50 pixels). We randomly held off a portion of the super-pixels (10, 30, or 50%) to be the test set while leaving the remaining super-pixels as the training set. Our goal is to evaluate whether CeLEry, Tangram, spaOTsc, and novoSpaRc can recover the 2D locations for the super-pixels in the test set.

We assessed the performance of CeLEry, Tangram, spaOTsc, and novoSpaRc using two approaches: (1) the Pearson correlation between the actual pairwise distance and the predicted pairwise distance for all cell pairs in the test data; and (2) the recovered 2D gene expression map using the predicted 2D coordinates. In general, CeLEry yielded a higher Pearson correlation between the actual and the predicted pairwise distances than Tangram, spaOTsc, and novoSpaRc (Fig. 5a and Supplementary Fig. 4a), especially when the holdoff proportion was high (Fig. 5b). Furthermore, the reconstructed 2D gene expression maps based on CeLEry's recovered 2D locations are more similar to the ground truth than those of Tangram, spaOTsc, and novoSpaRc (Fig. 5c) as indicated by higher scores for the structural similarity index measure (Supplementary Fig. 5).

To demonstrate the data augmentation step in CeLEry can improve the robustness of the prediction model, we artificially introduced Gaussian noise with varying degrees of standard errors into the gene expression data of the test set. As the degree of gene expression noise increases, the differences between the training and test data also increase, and the advantage of the augmented CeLEry model over the ordinary CeLEry model becomes more pronounced (Fig. 5d). This example demonstrates that the data augmentation procedure can enhance the model's ability to handle noise in the test data, thereby improving its robustness and generalizability.

We also examined the uncertainty estimates of the locations predicted by CeLEry. The uncertainty of the coordinate prediction varied spatially as the selected genes have different predictability in different regions. Typically, the certainty of the prediction tends to be higher for regions where the selected genes show enriched expression patterns. We used CeLEry to perform elliptical region prediction and measured the certainty of the prediction for each cell. Supplementary Fig. 6 shows that the certainty scores for spots were spatially variable, with the highest certainty score of -0.8 in the cerebellum region. The pattern for the region with relatively high certainty scores aligned well with the granular layer of the cerebellum.

In the above analyses, we used enhanced gene expression data obtained from TESLA as the spatial reference. The reason for doing this is that the original Visium data do not have a single-cell resolution. The ideal spatial reference is an ST dataset that covers the entire transcriptome and has a single-cell resolution. When such data are not available, we can use TESLA to enhance the spatial resolution of Visium. While this does not achieve a single-cell resolution, it offers a practical solution when only low-resolution ST data, such as Visium, are available. As shown in Supplementary Fig. 4b, when using the original Visium data as the spatial reference, the performance of all

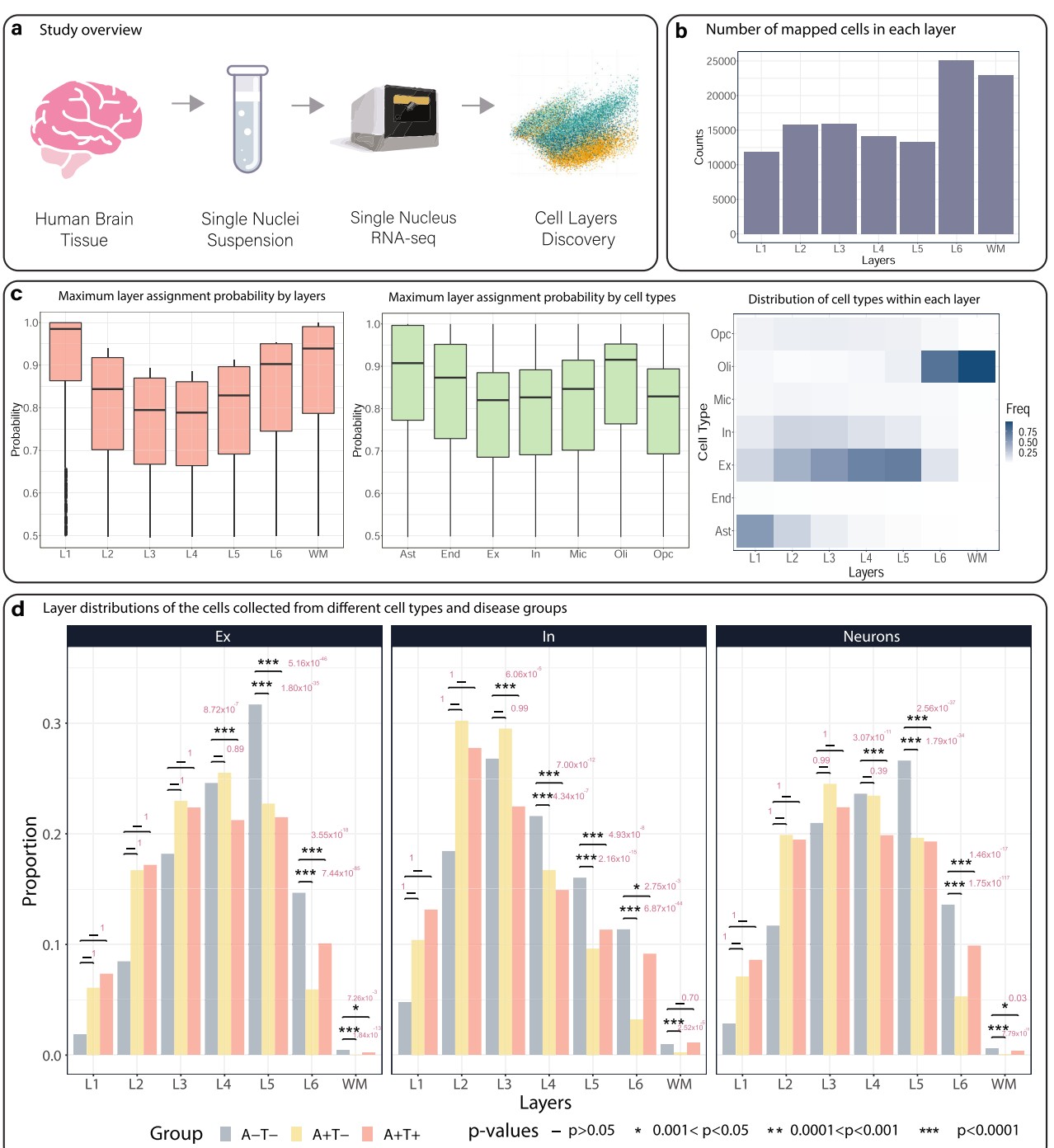

**Fig. 4 | Layer recovery for single cells in the AD study. a** Overview of the snRNA-seq data generation and analysis procedure for the AD study. In total, 15 post-mortem brains were processed for snRNA-seq and analysis. **b** Numbers of cells mapped to L1–L6 and WM by CeLEry. L1: layer 1, L2: layer 2, L3: layer 3, L4: layer 4, L5: layer 5, L6: layer 6, WM white matter. **c** Boxplots for the distribution of the maximum probability of assigning a cell to a layer (left panel; L1: $n = 11,807$ cells; L2: $n = 15,745$ cells; L3: $n = 15,949$ cells; L4: $n = 14,121$ cells; L5: $n = 13,332$ cells; L6: $n = 25,107$ cells; WM: $n = 22,988$ cells) and of assigning a cell to a layer by cell types (middle panel; Ast: $n = 12,126$ cells; End: $n = 444$ cells; Ex: $n = 39,176$ cells; In: $n = 12,286$ cells; Mic: $n = 3,982$ cells; Oli: $n = 44,182$ cells; Opc: $n = 6,853$ cells). In each boxplot, the lower and upper hinges correspond to the first and third quartiles, and the center refers to the median value. The upper (lower) whiskers extend from the hinge to the largest (smallest) value no further (at most) than the 1.5 × interquartile range from the hinge. Data beyond the end of the whiskers are plotted

individually. Layers or cell types that have higher mapping certainty will show higher maximum probabilities. Proportions of cells that are mapped to each layer by cell types (right panel). L1–L6 and WM were defined similarly as in Fig. 4b. Ast astrocyte, End endothelial, Ex excitatory neuron, In an inhibitory neuron, Mic microglia, Oli oligodendrocyte, Opc oligodendrocyte progenitor cell.
**d** Comparison of the proportions of neuronal cells mapped to each layer by disease status (left: excitatory neurons; middle: inhibitory neurons; right: excitatory and inhibitory neurons combined). Within each cell type, we compared the mapped cell proportions by layer between the A+T− and A−T− groups and between the A+T+ and A−T− groups. The comparison was conducted by a one-sided two-sample $Z$-test to test the null hypothesis that the diseased group (A+T− or A+T+) has more neuronal cells than the control group (A−T−). L1–L6 and WM were defined similarly as in Fig. 4b. Source data are provided as a Source Data file.

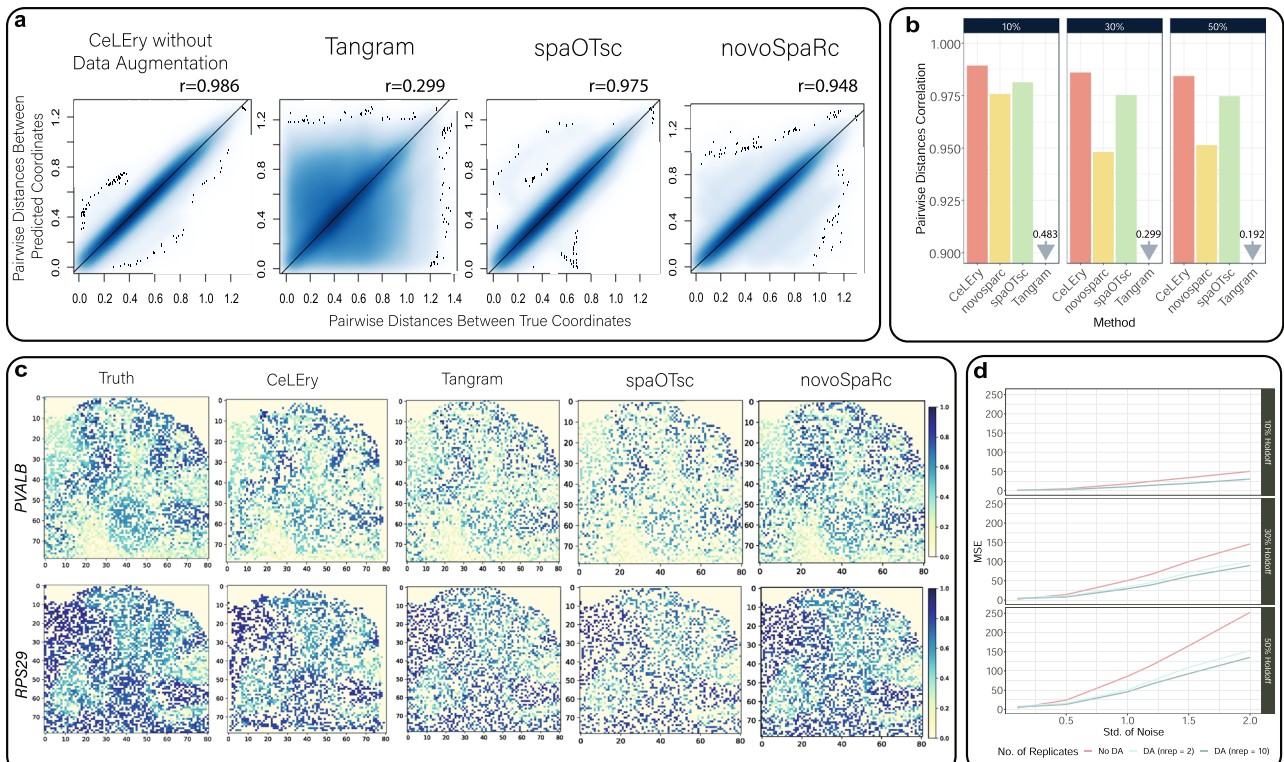

**Fig. 5 | 2D location recovery for spots in the mouse posterior brain data.**
**a** Scatter density plots of pairwise distances of true locations versus the predicted locations when the holdoff rate was 30% after resolution enhancement by TESLA. **b** Pearson's correlation of the pairwise distances of predicted locations and the distances of true locations for CeLEry, novoSpaRc, spaOTsc, and Tangram when the holdoff rate was set at 10, 30, and 50%, respectively. **c** The recovered gene

expression map in the test set based on predicted locations when the holdoff rate was 30%. The color shows relative gene expression. **d** MSEs of the predicted 2D locations when Gaussian noise with different standard deviations was added to the test data for scenarios with 10%, 30%, and 50% holdoff rates, respectively. The red line represents Tangram, and the shallow and dark blue lines represent CeLEry with two and ten replicates, respectively. Source data are provided as a Source Data file.

methods became worse; however, CeLEry still outperformed the other methods. In general, we recommend that users enhance the gene expression resolution first when single-cell resolution ST training data are not available.

## Benchmark evaluation for 2D location recovery in mouse brain with MERSCOPE data as spatial reference

While the previous results are encouraging, all of the analyses used data generated from 10x Visium as the spatial reference. Since Visium does not have a single-cell resolution, there is a possibility that the algorithm is "forced" to recover the spatial location information of single cells using the mixed spatial transcriptomics data as a reference. While the spatial resolution of the Visium data can be enhanced by programs such as TESLA, the resulting data still do not have a single-cell resolution. To eliminate this concern, we considered a situation where the spatial reference dataset is generated from a platform that has a single-cell resolution. Specifically, we analyzed a MERSCOPE dataset generated from a mouse brain by Vizgen[24]. For this dataset, we designed three scenarios, as summarized in Fig. 6a. Scenario 1 involved one replicate (sample ID: S1R1) containing 78,329 cells and 649 genes. After data quality control, we kept 18,342 cells with more than 100 expressed genes and 500 total UMI counts. We were stringent in filtering because MERSCOPE does not cover the entire transcriptome, and many of its measured genes were not expressed or were only expressed in a few cells. In this analysis, we split the data into two halves (left and right brain), using cells from the left brain for training and cells from the right brain for testing, and vice versa. This allowed us to recover the cell location information for the entire brain. Furthermore, since this MERSCOPE dataset includes two other replicates for the same slice (S1R2 with 88,884 cells and S1R3 with 84,636 cells), it

provided an opportunity to evaluate the performance of CeLEry in more complicated situations. Therefore, in Scenarios 2 and 3, we ensured that the training set and testing set were from different replicates. In Scenario 2, we used cells in the right brain of replicate S1R2 for training and tested on cells in S1R1. Since mouse brains are symmetric, it is rational to use one half of the brain to train the model and obtain predictions for both halves of the brain in the testing set. In Scenario 3, we merged the right brains of replicates S1R2 and S1R3, as their locations matched well, to enlarge the sample size of the training set and introduce more variable samples, and used the same testing set as Scenarios 2 (Fig. 6b).

We first examined whether CeLEry could retain the pairwise spatial distance between every two cells. We calculated the Pearson correlation between the true pairwise distance and the predicted pairwise distance for all cell pairs in this dataset. As shown in Fig. 6c, d, in Scenario 1, CeLEry achieved a correlation of 0.72, outperforming Tangram, spaOTsc, and novoSpaRc, whose had correlations of only 0.38, 0.67, and 0.56, respectively. In Scenario 2, where we applied the trained model to a different replicate, CeLEry still demonstrated robust performance with a correlation of 0.69. While spaOTsc also maintained stable prediction power with a correlation of 0.64, the correlations of Tangram and novoSpaRc dropped to 0.24 and 0.40, respectively, demonstrating that they are sensitive to variability between different replicates. When multiple training replicates were used (Scenario 3), CeLEry's performance improved further, achieving a correlation of 0.74, which is even higher than when the training set and testing set were from the same replicate (Scenario 1).

To better understand the accuracy of cell location recovery in different regions, we calculated the Euclidean distance between the true and predicted absolute locations for each cell in the test data and

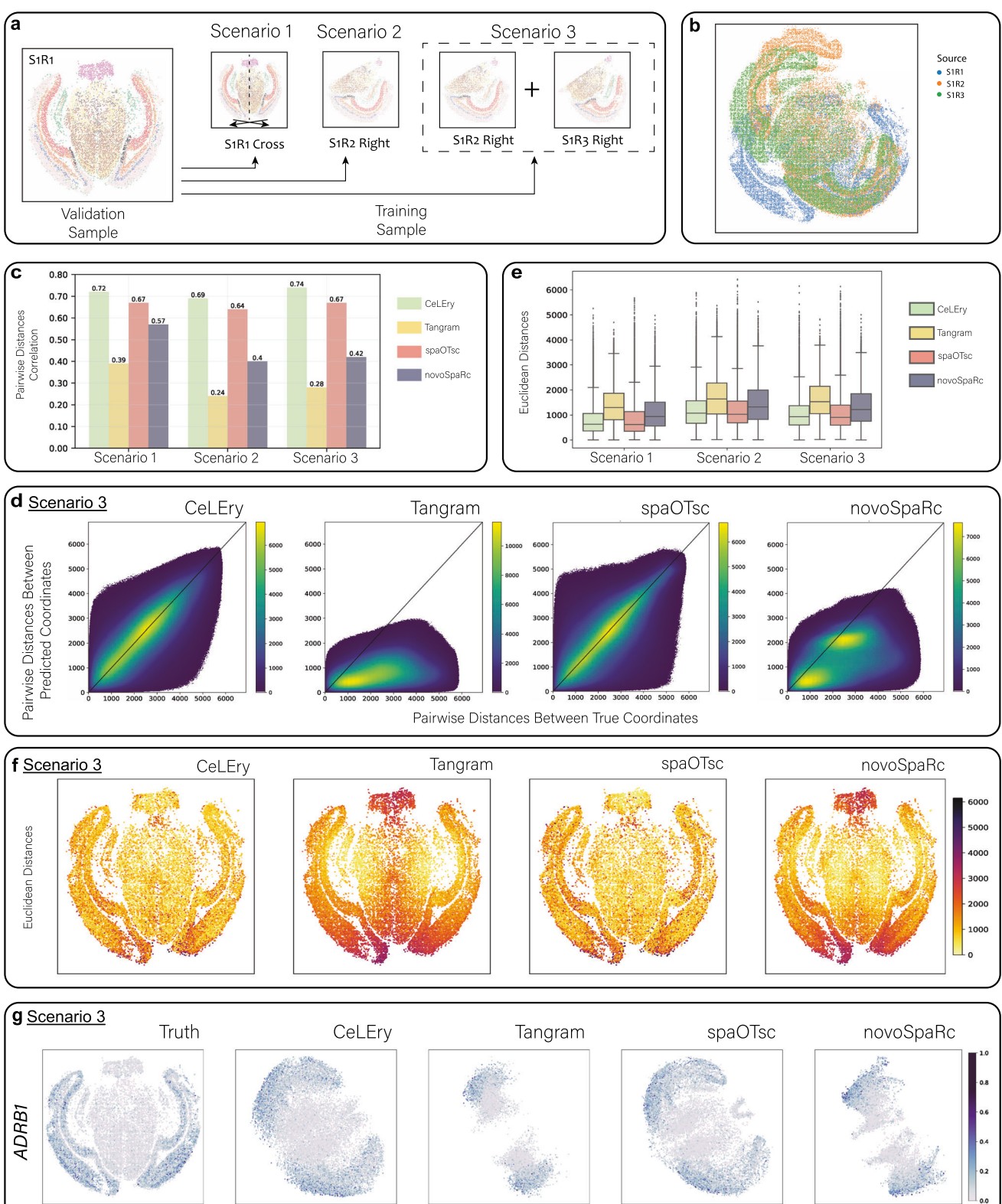

plotted the distribution as boxplots. However, in Scenarios 2 and 3, the predicted locations were in the space of the training replicate(s) and had a different orientation from the ground truth in Scenario 1. To make them comparable, we rotated the ground truth in Scenario 1 to roughly match their original orientation. Due to imperfect matching, it was not fair to directly compare Scenario 1 with Scenarios 2 and 3. However, we can still compare the performance of different methods within a single scenario or between Scenarios 2 and 3, which takes into account the effect of imperfect matching. The results presented in Fig. 6e, f and Supplementary Fig. 7 show that CeLEry outperformed Tangram and novoSpaRc, and was on par with spaOTsc in all scenarios. However, both CeLEry and spaOTsc struggled to map cells from similar regions. In Scenario 3, the Euclidean distances between predicted and true locations were generally lower than in Scenario 2, suggesting that increasing the training sample size improves prediction performance. Figure 6g and Supplementary Fig. 8 show randomly

**Fig. 6 | 2D location recovery for single cells in MERSCOPE mouse brain data.**
**a** Three scenarios with varying degrees of complexity were considered for benchmark evaluations. The color of each cell indicates cluster assignment obtained from unsupervised clustering. **b** Overlay of the three replicates based on their spatial coordinates. **c** Barplot of Pearson correlation between true and predicted pairwise distances for all cell pairs. **d** Scatter density plot comparing true and predicted pairwise distances for all pairs in Scenario 3. Color in the plot indicates the density of cell pairs. **e** Boxplot of Euclidean distances between true and predicted locations for all cells in the test data ($n = 18,342$ cells for each method under all scenarios). In each boxplot, the lower and upper hinges correspond to the first and third quartiles, and the center refers to the median value. The upper (lower) whiskers extend from the hinge to the largest (smallest) value no further (at most) than 1.5 × interquartile range from the hinge. Data beyond the end of the whiskers are plotted individually. **f** Visualization of Euclidean distances between true and predicted locations for all cells in the test data for Scenario 3. **g** Recovered gene expression map of a randomly selected gene *ADRB1*, based on the predicted locations by CeLEry in Scenario 3, with color indicating relative gene expression. Source data are provided as a Source Data file.

selected genes, whose spatial gene expression patterns were well recovered by CeLEry and spaOTsc. These analyses indicate that the location recovered single-cell data from CeLEry can recapitulate the original spatial expression patterns of the cells.

### Benchmark evaluation for 2D location recovery in mouse brain with MERFISH data as spatial reference

Motivated by the promising performance of CeLEry when using single-cell resolution ST reference in the mouse brain, we explored another single-cell resolution ST reference dataset that has more cells and genes. Specifically, we analyzed the Mouse3_sagittal dataset generated by Zhang et al.[25] using MERFISH. One advantage of this dataset is that it includes 1147 genes, which is more than the previous MERSCOPE dataset, providing more available features for location recovery. Additionally, it includes 25 coronal slices, allowing us to examine the scenario where the training and testing sets are from different tissue slices. For the sake of illustration, we selected three slices (Slice ID: sa1_slice4, sa1_slice5, sa1_slice7) with similar location regions and orientations to evaluate the performance of CeLEry. The data quality control criteria were the same as the study of mouse brain MERSCOPE data.

In Fig. 7a, we considered two scenarios for evaluating the performance of CeLEry on the Mouse3_sagittal dataset. In Scenario 1, we used cells in Slice 5 as the training and cells in Slice 7 as the testing set. In Scenario 2, we combined cells from Slice 5 and Slice 4 to form the training set and evaluated the performance of CeLEry on the same test data as Scenario 1. However, because Slice 4 and Slice 5 had some discrepancies in their location regions, we shifted the locations of Slice 4 to approximately align them with Slice 5, as shown in Fig. 7b. Due to the large number of cells in the dataset, spaOTsc and novoSpaRc were not able to handle Scenario 2. To evaluate the prediction accuracy, we compared the true pairwise distances with the predicted pairwise distances for all cell pairs and calculated the Pearson correlation. In Scenario 1, CeLEry reached a correlation of 0.89, outperforming Tangram (0.68), spaOTsc (0.81), and novoSpaRc (0.71) (Fig. 7c, d and Supplementary Fig. 9a). Expanding the training set in Scenario 2 led to improved accuracy of CeLEry, with a correlation of 0.90, while Tangram's correlation fell to 0.63. To investigate the spatial variability in prediction accuracy across spatial locations, we calculated the Euclidean distance between the true and predicted locations for each cell of the test data. As shown in Fig. 7e, CeLEry outperformed Tangram in both scenarios, with generally lower discrepancies between true and predicted locations. Note that spaOTsc and novoSpaRc were not able to be implemented in Scenario 2 due to the sample size of training data way exceeding their computation capacity. Furthermore, Fig. 7f and Supplementary Fig. 9b revealed that CeLEry maintained a consistent level of prediction accuracy across all locations, while Tangram, spaOTsc, and novoSpaRc exhibited spatial variability and performed poorly in some regions in both scenarios.

We conducted further validation of our spatial reconstruction results by analyzing cell-cell communication patterns. To retrieve known ligand-receptor pairs, we utilized CellChatDB[26], an existing database for storing ligand-receptor pairs. We selected several pairs whose gene expression showed spatial patterns and compared their expression patterns for ligands and receptors under true locations and

locations predicted by CeLEry. As depicted in Fig. 7g and Supplementary Fig. 9c, CeLEry was able to recover the expression patterns of the ligand and receptor pairs, with their expression exhibiting colocalized patterns in certain regions of the brain, suggesting potential cell-cell communications in those regions.

### Application to recover 2D locations for scRNA-seq data in mouse brain

To demonstrate the practical utility of CeLEry in predicting 2D cell locations, we applied it to a scRNA-seq dataset obtained from mouse brain[27]. For illustration purposes, we randomly sampled 500 cells from cortical layers, including L2 IT ENTl, L2/3 IT CTX, L4/5 IT CTX, L5 IT CTX, L6 CT CTX, and L6b CT ENT, resulting in a total of 3000 cells. Our objective was to assess the reliability of CeLEry in predicting the 2D locations of these cells. To evaluate the robustness of the model with different ST references, we conducted two studies: one using the MERSCOPE mouse brain data (sample ID: S1R1) generated by Vizgen as the reference and the other using Mouse2_Coronal dataset (slice ID: co2_slice80) from the MERFISH mouse brain data generated by ref. 25 as the reference. We trained the models on these reference data to learn the relationship between gene expression and 2D coordinates and then utilized the trained model to predict the locations of the sampled cells from the scRNA-seq data.

We would like to note that in our analysis with the MERFISH mouse brain data (ref. 25) as the spatial reference, spaOTsc and novoSpaRc were not able to process all of the cells in the training data. Therefore, we randomly sampled 30% and 40% of the cells, respectively, from the MERFISH mouse brain data as the reference for spaOTsc and novoSpaRc, during the training procedure. As shown in Fig. 8, the cell locations predicted by CeLEry revealed a clear cortical layer structure that aligned with the known cortical layer information of these cells, irrespective of the ST reference utilized. In contrast, the locations predicted by Tangram, spaOTsc, and novoSpaRc did not correspond well with the expected results, with many cells predicted to the wrong locations.

### Benchmark evaluation for 2D location recovery in liver cancer with MERSCOPE data as spatial reference

The development of CeLEry was initially motivated by the analysis of ST data from brain tissues. However, the framework of CeLEry is general and can be applied to analyze other tissue types. To examine CeLEry's performance in tissues other than the brain, we analyzed a liver cancer dataset generated using MERSCOPE[28], which consisted of 598,141 cells and 550 genes. We performed the same data quality control procedure as in the study of MERSCOPE mouse brain data, resulting in 159,074 cells for analysis. As the dataset only contained one replicate, we randomly held back 50% of the cells as training data, and the trained model was used to predict the locations of the remaining cells. The performance of different methods was evaluated by comparing the recovered cell maps to the original cell maps, as shown in Supplementary Fig. 10a. Notably, Tangram, SpaOTsc, and novoSpaRc were unable to process the testing data in its entirety as the sample size of the test data exceeds their computational capabilities. For the sake of comparison, a subset of 25% of the test cells was presented in Supplementary Fig. 10a. CeLEry demonstrated reasonable

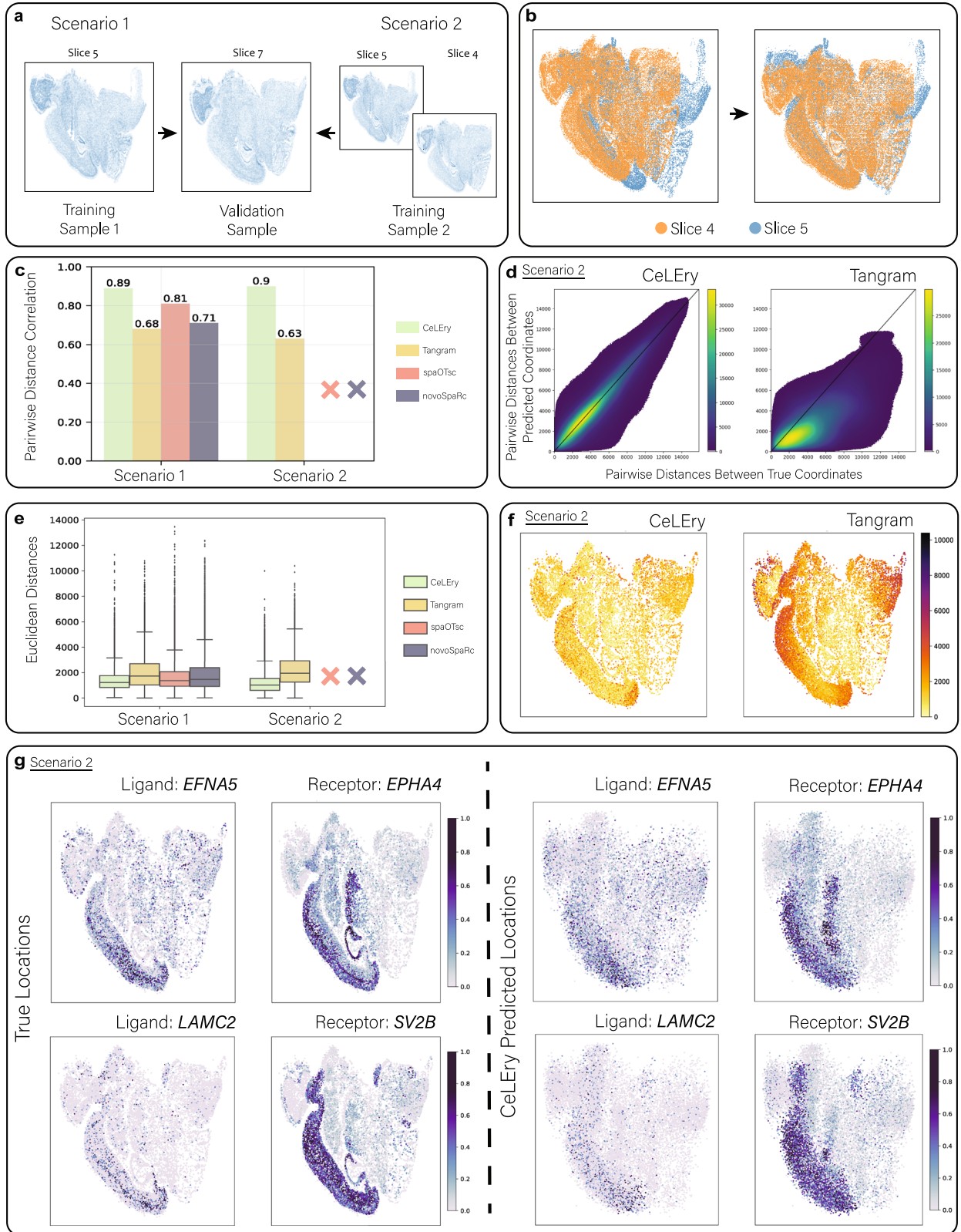

performance as the distribution pattern of cell types in the predicted cell map resembles the shape and structure of the original tissue. Meanwhile, Tangram and novoSpaRc both failed to recover the true spatial location, and spaOTsc failed to recover the top left regions. While recovering locations of cells from cancer tissue with a complex structure is challenging, CeLEry demonstrated reasonable performance as the distribution pattern of cell types in the predicted cell

map resemble the shape and structure of the original tissue, whereas Tangram and novoSpaRc both failed to recover the true spatial location and SpaOTsc failed to recover the top left regions.

The ability of CeLEry to retain the spatial pairwise distance of the cells was also evaluated by calculating the Pearson correlation between the true and predicted pairwise distance for all cell pairs in the dataset. As shown in Fig. 9a, b, CeLEry achieved a 0.68 correlation, surpassing

**Fig. 7 | 2D location recovery for single cells in MERFISH mouse brain data. a** Two scenarios with varying degrees of complexity were considered for benchmark evaluations. **b** Overlay of the two training slices based on their original spatial coordinates (left) and coordinates after manual location matching (right). **c** Barplot of Pearson correlation between true and predicted pairwise distances for all cell pairs. SpaOTsc and novoSpaRc were not able to handle a large number of cells in Scenario 2. **d** Scatter density plot comparing true and predicted pairwise distances for all cell pairs in Scenario 2. Color in the plot indicates the density of cell pairs. **e** Boxplot of Euclidean distances between true and predicted locations for all cells in the test data ($n = 18{,}197$ cells for each method under all scenarios). SpaOTsc and

novoSpaRc were not able to handle a large number of cells in Scenario 2. In each boxplot, the lower and upper hinges correspond to the first and third quartiles, and the center refers to the median value. The upper (lower) whiskers extend from the hinge to the largest (smallest) value no further (at most) than the 1.5 × interquartile range from the hinge. Data beyond the end of the whiskers are plotted individually. **f** Visualization of Euclidean distance between the true and predicted locations for each cell in the test data for Scenario 2. **g** True and recovered gene expression maps of two ligand-receptor pairs, based on the predicted locations by CeLEry, with color indicating relative gene expression for Scenario 2. Source data are provided as a Source Data file.

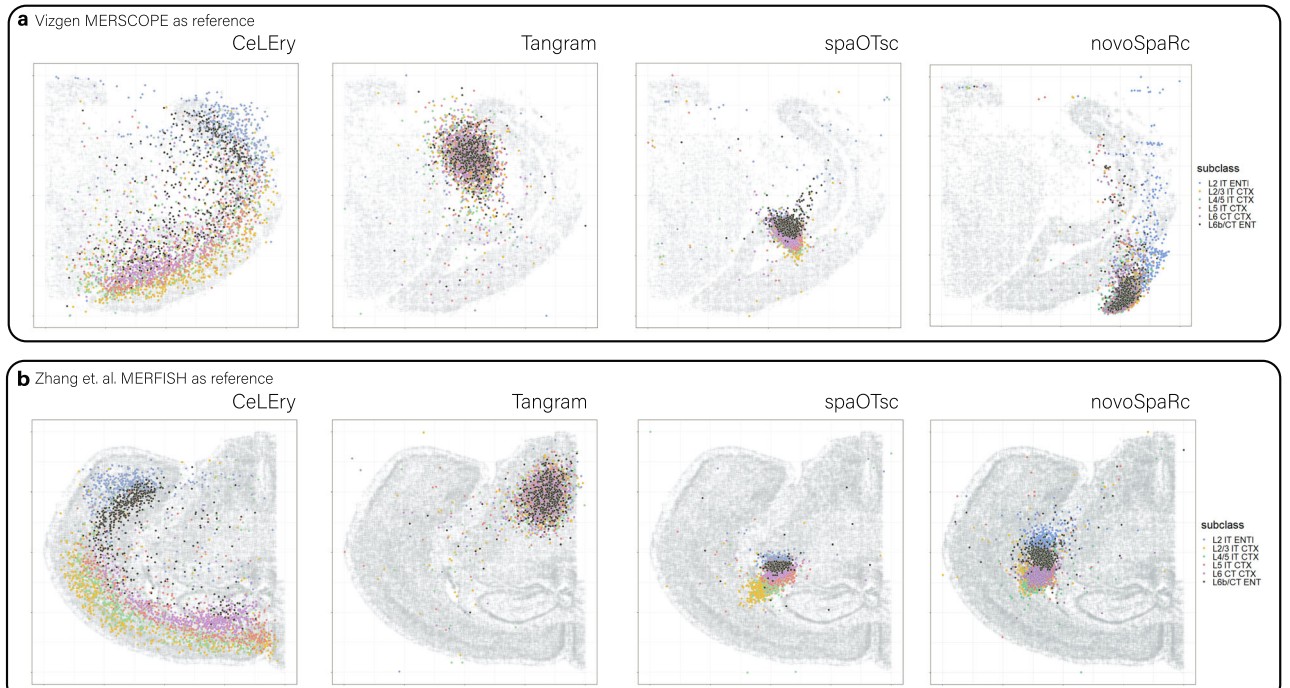

**Fig. 8 | Application to recover 2D locations for scRNA-seq data in mouse brain with different spatial references. a** 2D locations for scRNA-seq cells predicted by CeLEry, Tangram, spaOTsc, and novoSpaRc, using Vizgen's MERSCOPE mouse brain data as the spatial reference. **b** 2D locations for scRNA-seq cells predicted by

CeLEry, Tangram, spaOTsc, and novoSpaRc, using MERFISH mouse brain data generated by Zhang et al. as the spatial reference. Source data are provided as a Source Data file.

the performance of Tangram, spaOTsc, and novoSpaRc, which were 0.32, 0.37, and 0.33, respectively. Furthermore, we explored the spatial heterogeneity of the prediction accuracy by computing the Euclidean distance between the true and predicted locations for each cell of the test data. As shown in Fig. 9c, d, CeLEry consistently predicted the cell locations more accurately than Tangram, spaOTsc, and novoSpaRc, across all locations in the tissue. Finally, Fig. 9e and Supplementary Fig. 10b show randomly selected genes, whose spatial gene expression patterns can be roughly recovered by CeLEry using the predicted spatial locations of the cells.

**Benchmark evaluation for 2D location recovery in breast cancer with 10x Xenium data as spatial reference**

Building on the success of CeLEry in analyzing the MERSCOPE liver cancer data, we decided to investigate its applicability in another cancer dataset. Specifically, we analyzed a HER2-positive, ESR1-positive, and PGR-negative breast tumor dataset obtained from the recently released 10x Xenium platform[29]. This dataset contains information from two consecutively cut tissue sections from a breast cancer patient. Since the Xenium breast cancer data include two tissue sections, this allowed us to evaluate the performance of CeLEry when the training and test data are from different tissue sections. In our analysis, we utilized the slightly larger tissue section, *Replicate 1*, consisting of

167,782 cells for training the models and the smaller section, *Replicate 2*, consisting of 118,708 cells for validation. Both of the Xenium In Situ breast cancer tissue replicates contain data on a panel of 313 matching genes. Since this dataset is noisy and has a limited number of genes, we filtered out cells in *Replicate 2* that ranked in the bottom 75% for total UMI counts and the bottom 25% for the number of expressed genes for the 2D coordinate predictions. This resulted in a total of 29,770 cells in the validation set for our analysis. Because the competing methods were unable to handle the computational load of this task, we had to perform subsampling of the cells prior to running these methods. We ensured to maintain the same training-to-test size ratio that was used for CeLEry and subsampled 50,000 cells for both the novoSpaRc and Tangram analyses and 40,000 cells for the SpaOTsc analyses based on the computational capacity of the methods.

The Xenium in situ *replicate 1* training dataset is at single-cell resolution. However, in practice, users may not have access to a single-cell resolution dataset for model training for their specific question of interest. Therefore, we considered three schemes to simulate different training scenarios that a user may face when attempting to recover the 2D architecture of their data (Fig. 10a). The first scenario is the baseline where we used the original Xenium single-cell resolution dataset for model training. Next, for Scenario 2, we generated an artificial Visium dataset from the Xenium single-cell *replicate 1* dataset to represent a

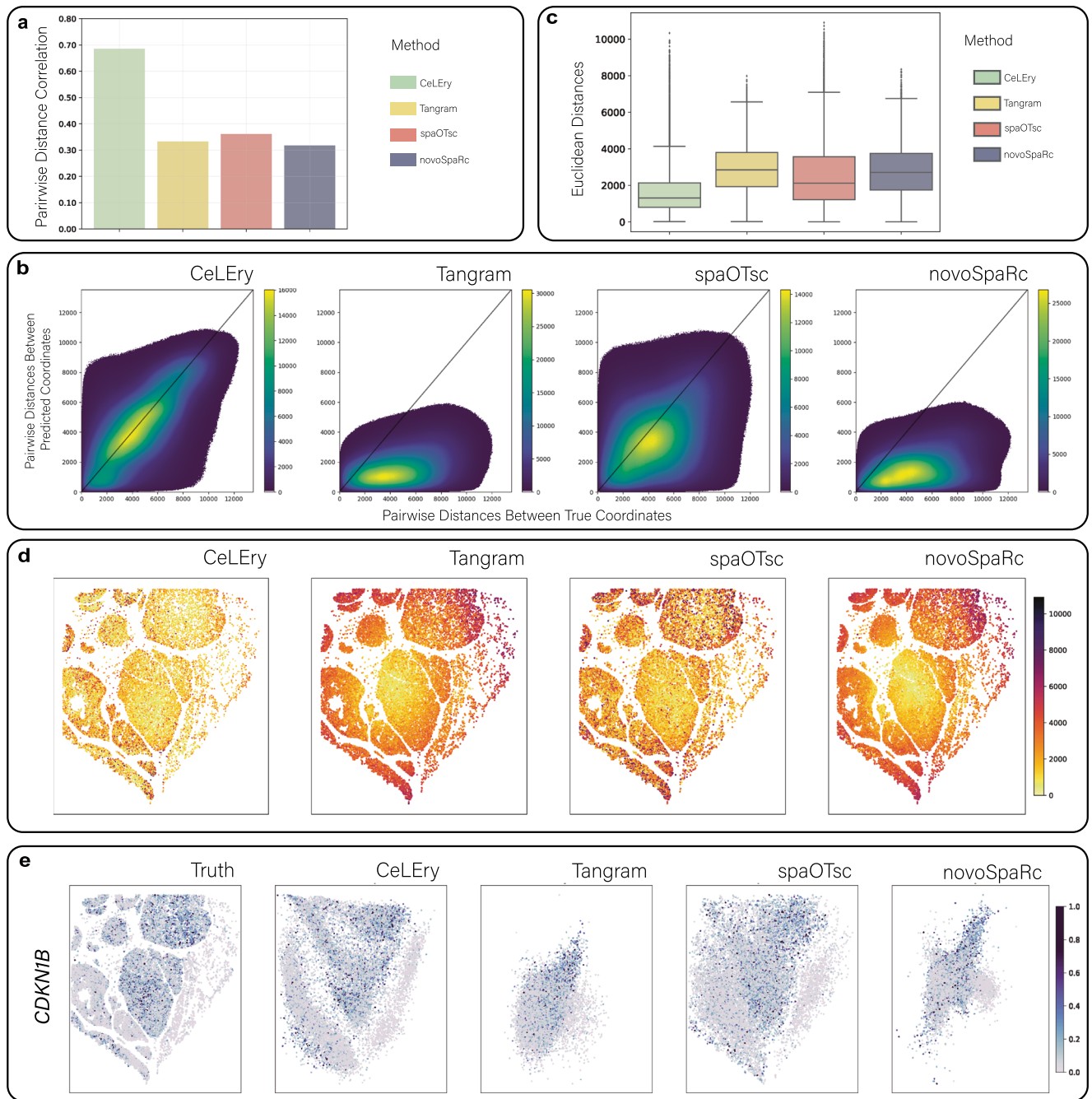

**Fig. 9 | 2D location recovery for single cells in MERSCOPE human liver cancer data. a** Barplot of Pearson correlation between true and predicted pairwise distances for all cell pairs. **b** Scatter density plot comparing true and predicted pairwise distances for all cell pairs. Color in the plot indicates the density of cell pairs. **c** Boxplot of Euclidean distances between true and predicted locations for all cells in the test data ($n = 19,885$ cells). In each boxplot, the lower and upper hinges correspond to the first and third quartiles, and the center refers to the median value. The upper (lower) whiskers extend from the hinge to the largest (smallest) value no further (at most) than the 1.5 × interquartile range from the hinge. Data beyond the end of the whiskers are plotted individually. **d** Visualization of Euclidean distance between the true and predicted locations for each cell in the test data. **e** Recovered gene expression map of a randomly selected gene *CDKN1B*, based on the predicted locations by CeLEry, with color indicating relative gene expression. Source data are provided as a Source Data file.

lower-resolution spot-level gene expression dataset containing 4215 spots, similar to what users may have access to and use for model training in practice. For Scenario 3, we enhanced the gene expression resolution of the artificial Visium training dataset from Scenario 2 using TESLA[23] in order to assess improvements in prediction performance from increasing the gene expression resolution when a histology image is available.

When assessing the performance of the methods across the three scenarios, we first analyzed the relationship between the true and predicted pairwise distances among each cell pair. The results, depicted in Fig. 10b and Supplementary Fig. 11a, show that CeLEry achieved significantly higher correlations of 0.74, 0.46, and 0.51 between the true and predicted pairwise distances across all three scenarios, respectively, compared to Tangram (0.33, 0.2, and 0.25), spaOTsc (0.41, 0.35, and 0.36), and novoSpaRc (0.43, 0.39, and 0.42). We also see the relationship between the true and predicted pairwise distances from CeLEry much more closely follow a diagonal 45-degree line in all three scenarios compared to the other competing methods,

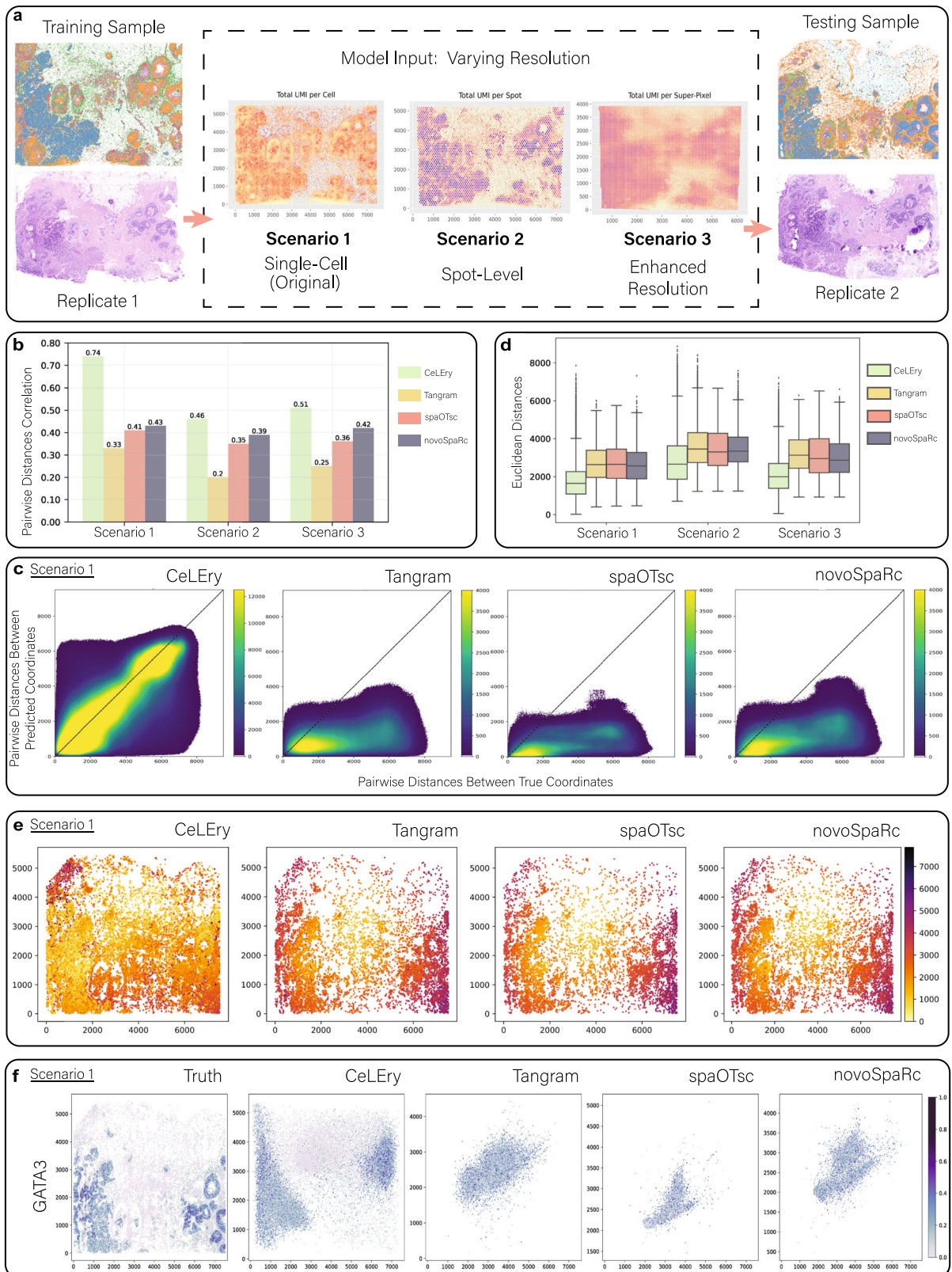

as shown in Fig. 10c and Supplementary Fig. 12a, c. Here we see using the original single-cell resolution Xenium data for model training results in the highest pairwise correlation for all methods (Scenario 1). The results show an improvement in the reconstruction of pairwise distances when using the TESLA-enhanced gene expression across all methods (Scenario 3) compared to the artificial Visium spot-level resolution data (Scenario 2).

To further explore the spatial heterogeneity of the prediction accuracy, we calculated the Euclidean distance between the true and predicted locations of each cell in the test data for each of the proposed scenarios. The results, shown in Fig. 10d, indicate that coordinates recovered by CeLEry have an overall lower Euclidean distance from the true coordinates compared to the coordinates recovered by Tangram, spaOTsc, and novoSpaRc. The median Euclidean distance

**Fig. 10 | 2D location recovery for single cells in 10X Xenium breast cancer data.** **a** Visualization of Replicate 1 and the three training scenarios investigated with varying spatial resolutions, including Xenium single-cell, artificial Visium spot-level, and enhanced spot-level, along with visualization of the single-cell Xenium replicate 2 test dataset. The color of each cell in the training sample and testing sample indicates cluster assignment obtained from unsupervised clustering. **b** Side-by-side bar graph of Pearson correlation between true and predicted pairwise distances across all cell pairs for each method and scenario evaluated. **c** Scatter density plots comparing true and predicted pairwise distances for all cell pairs in Scenario 1. Color in the plot indicates the density of cell pairs. **d** Side-by-side boxplots of Euclidean distances between true and predicted locations for all cells in the test data for each method and scenario evaluated (Scenario 1: *n* cells in the test set = 29,770, 8872, 7097, and 8872 for CeLEry, Tangram, SpaOTsc, and novoSpaRc

respectively due to the computational capacity of each method; Scenario 2: *n* cells in the test set = 29,770 for each method; Scenario 3: *n* cells in the test set = 29,770, 8872, 7097, and 8872 for CeLEry, Tangram, SpaOTsc, and novoSpaRc respectively due to the computational capacity of each method. In each boxplot, the lower and upper hinges correspond to the first and third quartiles, and the center refers to the median value. The upper (lower) whiskers extend from the hinge to the largest (smallest) value no further than (at most) than 1.5 × interquartile range from the hinge. Data beyond the end of the whiskers are plotted individually. **e** Visualization of Euclidean distances between true and predicted locations for all cells in the test data for Scenario 1. **f** Recovered gene expression map for Scenario 1 for a randomly selected gene, *GATA3*, with color indicating relative gene expression. Source data are provided as a Source Data file.

---

between the true and predicted coordinates from CeLEry were 1,645, 2,659, and 1995 across the three scenarios, respectively. The median Euclidean distance between the true and predicted coordinates were considerably higher for Tangram (2632, 3458, and 3133), spaOTsc (2651, 3302, and 2948), and novoSpaRc (2564, 3349, and 2863) for all three scenarios. The results indicate that using the single-cell Xenium In Situ data for model training results in predictions with the lowest Euclidean distance from the truth (Fig. 10d). When comparing Scenarios 2 and 3, the results show that performing gene resolution enhancement on the Visium spot-level data leads to improvement in overall prediction with predicted locations being closer to the true locations across all methods. We note that although CeLEry does tend to have a larger range in the Euclidean distances compared to the other methods, we have found Tangram, spaOTsc, and novoSpaRc are all prone to predicting cells locations very close to the tissue center resulting in smaller bounds in the estimates. As shown in Fig. 10e and Supplementary Fig. 12b, d, CeLEry consistently performed well across different locations in the tissue, whereas Tangram, spaOTsc, and novoSpaRc, produced larger errors when predicting cells located far from the tissue center for all scenarios investigated. Furthermore, Fig. 10f and Supplementary Fig. 13 depict the recovered spatial gene expression patterns for selected genes showing various spatial patterns from the Xenium panel, including breast cancer genes *ERBB2* and *ESR1* for biological interest since the tissue section is classified as HER2 and ER-positive. Additionally, the tissue structure shown in Fig. 10a demonstrates a higher degree of cell heterogeneity on the x-axis than on the y-axis. Therefore, we see that it was generally easier to predict the x-axis than the y-axis, as the correlation of truth and predicted coordinates on the x-axis was higher than that of the y-axis (Supplementary Fig. 11b) for this type of tissue architecture. Overall, we show that CeLEry is able to successfully reconstruct the tissue architecture when training and testing from two different tissue sections. The results, as shown in Fig. 10a–c, indicate that using single-cell resolution data (Scenario 1), when available, for model training outperforms using Visium spot-level or spot-level enhanced data, as expected. However, when single-cell resolution data is not available for training, we have found that utilizing a gene resolution enhancing algorithm, such as TESLA[23], on the training data prior to predicting the 2D location coordinates results in better tissue reconstruction compared to training on Visium spot-level data for all methods evaluated.

## Discussion

In this paper, we presented CeLEry, a machine-learning method that recovers the spatial location information for cells in scRNA-seq data by leveraging information learned from ST. CeLEry employs a feedforward deep neural network to build a cell location prediction model, which can also include an optional data augmentation step based on a variational autoencoder to generate ST data replicates. The data augmentation step can increase the size of the training sample and improve performance when training samples are limited, and test data are noisy. CeLEry is capable of recovering both spatial domains and 2D

location information. Through extensive benchmarking evaluations on datasets covering a wide range of scenarios, we demonstrate that CeLEry outperformed other state-of-the-art methods such as Tangram, spaOTsc, and novoSpaRc for both spatial domain and 2D location recovery. These findings suggest that CeLEry is a powerful tool for analyzing scRNA-seq data and can contribute to a better understanding of cellular organization in tissues.

The spatial location recovered scRNA-seq data from CeLEry can be utilized in two different ways. First, these data can be utilized in scRNA-seq-centric analysis, such as cell-cell communications (CCC). CCC is an essential feature of multicellular organisms[30,31]. The binding of ligands to their corresponding receptors, which activate specific signaling pathways, is intimately linked to many complex human diseases. Since CCC is spatially coordinated[31], knowing the cellular localizations is crucial to understand how different cells interact with each other during disease development and progression. Our methods will enable CCC analysis for the large amount of data generated in big consortia such as HuBMAP[32] and the Human Cell Atlas[33]. Second, the spatially mapped scRNA-seq cells can also facilitate ST-centric analysis. For example, they can help infer cell-type composition and cell-type-specific gene expression in 10x Visium. Moreover, they can aid in imputing missing gene expression for genes that are not included in MERSCOPE, Xenium, and CosMx[34]. Such imputation will provide data generated from these platforms with full-transcriptome coverage and enable CCC analysis at the single-cell level.

It is important to note that most existing methods primarily focus on ST-centric analysis. These methods aim to improve the quality of ST by utilizing scRNA-seq data as a helper. As a result, accurately recovering the exact 2D location for each scRNA-seq cell is not their primary objective. Instead, the primary goal of these methods is to map a scRNA-seq cell to a specific tissue location that has the same cell type. As each cell type typically consists of multiple cells, mapping a cell from that cell type to the corresponding location is considered a correct mapping. This is a much simpler task methodologically compared to accurately mapping a scRNA-seq cell to its precise tissue location.

Although CeLEry was primarily developed for the purpose of recovering cell locations, its framework is flexible enough to be adapted for cell type mapping of the ST data, where the cell type label information in a well-annotated scRNA-seq dataset can be transferred to the unannotated ST data. To demonstrate this capability, we trained the spatial domain prediction model by treating each cell type as a spatial domain. This adaptation is reasonable as cell types do not possess an ordinal relationship. We conducted additional analysis on Replicate 1 of the 10x Xenium breast cancer data and found that when evaluating the performance of CeLEry by cell type mapping, the accuracy is substantially improved to 0.96 (Supplementary Fig. 14), outperforming Tangram, spaOTsc, and novoSpaRc. This result shows that CeLEry is versatile in its application.

While our real data analyses demonstrate that CeLEry has an outstanding performance in recovering the 2D coordinates, this task is

generally much more challenging than recovering spatial domains or cortical layers. This is because precise recovery of 2D coordinates relies on genes that exhibit fine-grained spatial gradient gene expression patterns in tissues, and it is not yet clear how many such genes exist. Therefore, identifying genes or sets of genes that show gradient expression patterns in tissues requires further investigation. In future studies, we will explore the identification of such gradient patterns by finding metagenes that combine the information from multiple genes. This will allow us to further improve the performance of 2D coordinate recovery in CeLEry.

In this study, we demonstrated two scenarios where data augmentation is useful. The first scenario is when the training model is prone to overfitting as the test data can be from a different individual or even a different species. The second scenario is when the test data are noisy, which is common for scRNA-seq. The choice of the replicate number produced by data augmentation affects the predictions. The number of replicates to be generated is a hyperparameter prespecified by users. Our results show that introducing augmented ST data from the generative model may improve the predictions, but if the number of replicates is too large, the prediction accuracy might be reduced. There is a balance between the original ST data and the augmented ST data. Ideally, the replication number should be chosen such that information from the two data sources is balanced. In our analyses, we found that having two replicates generally achieved the best performance. As shown in the application studies, choosing a larger number of replicates does not guarantee a higher prediction accuracy. As the number becomes larger, more proportion of training data will come from an artificial generation, and hence the model gives smaller weight to the original data. In real applications, we recommend that the user explore the best number of replicates by performing cross-validation. Our data augmentation procedure is optional and was designed for ST data that have a grid layout, e.g., 10x Visium. ST data with non-grid layout, e.g., SLIDE-seq, MERSCOPE, MERFISH, or 10x Xenium, would not benefit from this procedure. However, since the number of cells in MERSCOPE, MERFISH, and Xenium data is typically large, data augmentation is unnecessary.

The ST data generated from the 10x Visium platform lacks single-cell resolution, which leads to heterogeneity between the ST and scRNA-seq data. Despite using imperfect ST training data, CeLEry achieved a reasonable performance and outperformed Tangram, spaOTsc, and novoSpaRc in all evaluations. Moreover, our proposed data augmentation further improved CeLEry's performance. When applied to single-cell resolution ST data such as MERSCOPE, MERFISH, and Xenium, CeLEry demonstrated excellent performance and outperformed other competing methods. However, it is worth noting that MERSCOPE, MERFISH, and Xenium only measure a small number of genes. Therefore, since their gene panels may not include sufficient spatially variable genes, the performance of CeLEry is currently suboptimal. We believe CeLEry will perform even better when ST data with single-cell resolution and transcriptome-wide coverage become more widely available.

## Methods
Our method takes two datasets as input, a ST dataset as the reference to train a location prediction model, and a scRNA-seq (or snRNA-seq) dataset where the cell locations are unknown and are the primary target of our prediction. We consider both 2D location recovery and spatial domain recovery.

### Data preprocessing
The gene expression normalization involves two steps. In the first step, cell/spot-level normalization is performed in which a logarithm transformation is applied to the unique molecular identifier (UMI) count for each gene in each cell/spot. In the second step, gene-level normalization is performed in which the values for each gene are standardized by subtracting the mean across all cells/spots and dividing by the standard deviation across all cells/spots for the given gene. Spatially variable genes are selected using the rank_genes_groups function in the spaGCN package (for 2D location recovery), and differentially expressed genes are selected using the rank_genes_groups function in the Scanpy package (for spatial domain recovery). For the analysis of cortical layers in the LIBD data that involve multiple training tissue sections, we first perform gene expression normalization and logarithm transformation for each training section separately. Then, we subset the common genes shared by all sections and merge the gene expression and layer information to generate a combined training dataset.

### Data augmentation for the training ST data
Since ST data are expensive to generate, the training data might have a limited sample size. Thus, they are vulnerable to noise in the data and prone to overfitting. To overcome these challenges, we propose a data augmentation procedure to generate "replicates" of the original ST data. This procedure can be skipped if enough training data are available.

The data augmentation procedure has three steps. In Step 1, we transform the ST data by treating each gene as a sample. Let $Z \in R^{n \times p}$ be the gene expression matrix obtained from a ST dataset in which rows represent spots and columns represent genes, $n$ is the number of spots, and $p$ is the number of genes. For gene $j$, we rearrange its 1D expression vector $Z_j$ into a 2D matrix $X^j \in R^{n_1 \times n_2}$ according to the spatial coordinates of each spot, where $n_1$ is the total number of rows and $n_2$ is the total number of columns in the tissue capture area of the ST data. If a spot in $X^j$ does not have a corresponding spot in $Z_j$, its gene expression will be coded as 0.

In Step 2, we perform K-means clustering to cluster the genes into groups sharing similar expression patterns. The clustering results are encoded to a one-hot matrix.

In Step 3, we construct a variational autoencoder, which is initialized with a combination of an encoder and a decoder. The encoder starts with an input layer, which is of the same dimension as $X^j$, and the output is the mean and standard error parameters to be learned. All intermittent bottleneck layers are connected by convolutional neural networks, except that the last bottleneck layer and output layer are connected by a fully connected neural network. The decoder is composed of a generated embedding layer, an output layer, and a few middle bottleneck layers. The embedding layer is generated by a normal distribution with the mean and standard deviation given by the output layer of the encoder and then is concatenated with an indicator vector in the one-hot clustering results of each gene. The output layer is of the same dimension as $X^j$ and is trained to reconstruct the input layer. The parameters of the variation autoencoder are iteratively updated to minimize the following loss function:

$$Loss = \sum_{i \in S} \sum_{j=1}^{p} \left( X_i^j - \widehat{X}_i^j \right)^2 + \beta KL\left(N(\mu^j, \sigma^{j2}) | N(0,1)\right), \quad (1)$$

where $X_i^j$ is the gene expression for spot $i$ and $S$ is a set that records the spatial coordinates of all spots. The first component in the above loss function represents the reconstruction loss, the second component is the Kullback–Leibler (KL) divergence that characterizes the similarity between the distribution of the generated embedding and the standard normal distribution, and $\beta$ is a tuning parameter that controls the weight between the loss components. The value of $\beta$ is set at $10^{-5}$ by default to make the loss component of similar magnitude. The KL divergence is computed as

$$KL\left(N(\mu^j, \sigma^{j2}) | N(0,1)\right) = \frac{1}{2} \sum_{d=1}^{D} \left( \sigma_d^{j2} + \mu_d^{j2} - 1 - \log\left(\sigma_d^{j2}\right) \right), \quad (2)$$

where $D$ is the dimension of the embedding layer and is set at 512 by default.

After convergence of the model, we use the trained encoder to estimate $\mu_j$ and $\sigma_j^2$ for each gene and generate a series of embeddings from $N(\mu^j, \sigma^{j2})$. Then, for each generated embedding, we apply the decoder to reconstruct the spatial gene expression matrix, which will be combined with the original ST data for model training in the location prediction model described below.

## Location prediction model

Let $Z_i.$ be the gene expression of spot $i$ in $Z$. We train a supervised deep neural network to learn the relationship between gene expression features and the spatial locations in the ST dataset. The deep neural network is initialized with an input layer with the same dimension as $Z_i$, an output layer, and multiple latent layers in between whose dimensions decrease as the layers approach the output layer. Depending on specific research interests, the structure of the output layer and the loss function can be different. If the goal is to predict the 2D coordinates and its associated prediction region, the dimension of the output layer is 2, representing the spatial coordinates of the spot, denoted as $(\hat{y}_{i1}, \hat{y}_{i2})$. The parameters are updated using gradient descent to minimize the following loss function,

$$Loss = \sum_{i=1}^{n} \left[ (y_{i1} - \hat{y}_{i1})^2 + (y_{i2} - \hat{y}_{i2})^2 \right], \qquad (3)$$

where $(y_{i1}, y_{i2})$ are the true coordinates of spot $i$. To obtain the elliptical prediction region, we modify the last layer to be of 4 dimensions, i.e., $(\hat{c}_{i1}, \hat{c}_{i2}, \hat{r}_{i1}, \hat{r}_{i2})$ with $\hat{r}_{i1}$ and $\hat{r}_{i2}$ constrained to be positive. Let $s_i = I([(y_{i1} - \hat{c}_{i1})/\hat{r}_{i1}]^2 + [(y_{i2} - \hat{c}_{i2})/\hat{r}_{i2}]^2 \leq 1)$. The loss function of the uncertainty region for the predicted location is governed by axes $(\hat{r}_{i1}, \hat{r}_{i2})$ with coverage probability $\alpha$ given by

$$Loss = \sum_{i=1}^{n} (\alpha(1 - s_i) + (1 - \alpha)s_i) \left| \left( \frac{y_{i1} - \hat{c}_{i1}}{\hat{r}_{i1}} \right)^2 + \left( \frac{y_{i2} - \hat{c}_{i2}}{\hat{r}_{i2}} \right)^2 - 1 \right|, \qquad (4)$$

where $(\hat{c}_{i1}, \hat{c}_{i2})$ is the centroid of the confidence elliptical region.

To recover the spatial domain of a cell, we consider a categorical output, $\hat{a}_i$, in the deep neural network. We consider a logistic regression loss given by

$$Loss = -\sum_{i=1}^{n} \sum_{c=1}^{C} \left[ \log\{\sigma(\hat{a}_i)\} + \log\{1 - \sigma(\hat{a}_i)\} \right], \qquad (5)$$

where $\sigma(x_i) = \exp(x_i)/[1 + \exp(x_i)]$ and $C$ is the number of spatial domains. For spatial domains with orders, e.g., cortical layers in the brain, we consider a rank-consistent logistic regression loss given by

$$\begin{aligned} Loss = -\sum_{i=1}^{n} \sum_{l=1}^{L-1} [&\log\{\sigma(\hat{a}_i + b_l)\}I(y_i > l) \\ &+ \log\{1 - \sigma(\hat{a}_i + b_l)\}\{1 - I(y_i > l)\}], \end{aligned} \qquad (6)$$

where $y_i$ is the layer index that spot $i$ belongs to, $b_l$ is the cut point parameter between layer $l$ and layer $l + 1$, and $L$ is the total number of layers.

CeLEry allows users to specify the numbers of hidden layers and nodes. The default number of layers is 3, and the numbers of nodes are 50, 10, and 5 for the three layers, respectively. We suggest setting more hidden layers and more nodes per layer if the training sample size is large so that the complex relationship can be better learned through the deep neural network. We use ReLU as the activation function for the hidden layers. For the output layer, we use Sigmoid as the activation function in order to scale the output into the [0, 1] range. Rescaling 2D coordinates into the [0, 1] range will facilitate the comparison of results across different datasets. For layer (or spatial domain) recovery, the [0, 1]-ranged output represents the probability of assigning an observation to a layer (or spatial domain). The default variational autoencoder is constructed with three hidden 2D convolutional layers in the encoder, where each layer has 16, 8, and 4 channels. The decoder component has the same shape as the encoder but in a reversed order. i.e., the number of channels for each layer is 4, 8, and 16, respectively.

## Statistics and reproducibility
No statistical method was used to predetermine the sample size. Highly variable genes or top differentially expressed genes between spatial domains were selected when training the cell location prediction model. In the study of MERSCOPE, MERFISH, and 10X Xenium data, cells with low UMI counts or a small number of expressed genes were eliminated. The experiments were not randomized. The Investigators were not blinded to allocation during experiments and outcome assessment.

## Reporting summary
Further information on research design is available in the Nature Portfolio Reporting Summary linked to this article.

## Data availability

(1) "LIBD human DLPFC 10x Visium data [http://research.libd.org/spatialLIBD/]"; (2) "Mouse posterior brain 10x Visium data [https://support.10xgenomics.com/spatial-gene-expression/datasets/1.0.0/V1_Mouse_Brain_Sagittal_Posterior]"; (3) "AD snRNA-seq data [https://upenn.app.box.com/s/e8nf4b384s7oi3o09pj5s8jfdu11swim]"; (4) "Mouse brain MERSCOPE data [https://info.vizgen.com/mouse-brain-data]"; (5) "Mouse Brain MERFISH data [https://doi.org/10.35077/act-bag]"; (6) "Mouse Whole Cortex and Hippocampus 10x scRNA-seq data used to predict the 2D locations onto the mouse brain reference data [https://portal.brain-map.org/atlases-and-data/rnaseq/mouse-whole-cortex-and-hippocampus-10x]"; (7) "Human liver cancer MERSCOPE data [https://console.cloud.google.com/storage/browser/vz-ffpe-showcase/HumanLiverCancerPatient2]"; (8) "10X Xenium data [https://www.10xgenomics.com/welcome?closeUrl=%2F&lastTouchOfferName=Xenium+Preprint+Dataset&lastTouchOfferType=Dataset&product=xenium&redirectUrl=%2Fproducts%2Fxenium-in-situ%2Fpreview-dataset-human-breast]" are all publicly available. All other data supporting the findings of this study are available within the article and its supplementary files. Any additional requests for information can be directed to, and will be fulfilled by, the lead contact. Source data are provided with this paper.

## Code availability

An open-source implementation of the CeLEry algorithm can be downloaded from https://github.com/QihuangZhang/CeLEry. The codes are also available via Zenodo at https://doi.org/10.5281/zenodo.8019107[35].

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

## Acknowledgements
This work was supported by grants: R01GM125301, R01EY030192, R01EY031209, and R01HL150359 (to M.L.), and P01AG066597 (to M.L. and E.B.L.).

## Author contributions
This study was conceived of and led by M.L. Q.Z. designed the model and algorithm. Q.Z. implemented the CeLEry software and led data analyses for the LIBD data, the Alzheimer's disease snRNA-seq data, the 10x Visium mouse brain data, and the mouse brain scRNA-seq data. S.J. led data analyses for the MERSCOPE and MERFISH mouse brain data and MERSCOPE liver cancer data. A.S. led data analyses for the 10x Xenium breast cancer data. Q.Z., S.J., and A.S. analyzed the data with input from M.L., R.X., J.H., K.L., B.Z., D.D., and E.B.L. D.D. and E.B.L. provided feedback for the Alzheimer's disease snRNA-seq data analysis. Q.Z., M.L., S.J., and A.S. wrote the paper with feedback from all other co-authors.

## Competing interests
K.L. and B.Z. are employees of Biogen Inc. M.L. received research funding from Biogen Inc. unrelated to the current manuscript. The remaining authors declare no competing interests.
