## [Peer review file · Nature Communications]

REVIEWER COMMENTS

Reviewer #1 (Remarks to the Author):

In this manuscript, Zhang et al developed a computational method, CeLEry, to recover the location information for cells in scRNA-seq data by utilizing the gene expression and spatial location relationships learned from spatial transcriptomic data. They also proposed a data augmentation method that utilizes a VAE model to generate realistic replicates of the original spatial transcriptomic data, to increase the sample size of the training data to learn more relationships between the gene expression and spatial location in spatial transcriptomic data. They mainly compared the performance of CeLEry with Tangram on both spatial transcriptomic data and scRNA-seq data, and finally, they applied CeLEry to an AD snRNA-seq data to recover cortical layers information of cells. Since a large amount of scRNA-seq data have been generated, reconstructing spatial location information of cells is meaningful. However, I think there is still some further validation and expansion needed to make the significance of the CeLEry clearer. My main concerns are listed below, hoping these are useful to the authors.

Major:

1. One key concern of mine here relates to the theoretical justification of this method. The method trains the model on spatial transcriptomic data and then utilizes the trained model for scRNA-seq data. However, these (spatial transcriptomic data and scRNA-seq data) represent two different data types. The spatial transcriptomic data used in the manuscript is 10x Visium, it doesn't reach the single-cell resolution and each spot contains cells with different cell types. It seems a bit "forced" to reconstruct spatial location information of single cells using the mixed spatial transcriptomic data as a reference. Some validation experiments should be performed (perhaps comparing the spatial reconstruction results of scRNA-seq data of this method with deconvolution results of spatial transcriptomic data reference is a way to try).

2. All the validation and applications in the manuscript are based on the same set of spatial transcriptomic data (DLPFC). I would have to see results showing a good spatial reconstruction of different tissues to determine that this method is well scalable.

3. Although the idea of this manuscript is to recover cell locations of single cells, the authors mainly performed experiments that recover the spatial domain of cells. It's more like a classification task. In my opinion, recovering cell locations of single cells is more meaningful, but the authors only did a validation experiment of this in mouse brain data. I would like to see more results from the reconstruction of cell locations. The authors could create two long vectors (one for the reconstruction result and one for the ground truth) containing the distance between every pair of cells and then calculate the correlation between them to evaluate the performance of their method.

4. I appreciate that the authors benchmark their method, but would argue that their choice of method (Tangram) to benchmark against is a bit surprising. From my perspective, I think CeLEry is more like a method for spatial reconstruction of single cells. To make the comparison more relevant I would encourage the authors to benchmark their method with other spatial reconstruction methods, for example, novoSpaRc, SpaOTsc, etc.

5. In the data augmentation section, if I understand correctly, the authors created 2D gene expression graphs (like "photos") based on the spatial location and gene expression of spatial transcriptomic data and utilized a CNN network to extract the low dimensional feature representations. However, for the image-based spatial transcriptomic data like slide-seq, STARmap, or MERFISH, due to their non-Euclidean structure spatial locations, I don't know if this method still work? If not, the limitations should be addressed in the discussion.

6. In Figure 2b, the augmentation results of CALB1 don't seem very good. Does the use of these "realistic replicates" introduce additional noise? Besides, is there any particular reason for only selecting these two genes rather than the six gene clusters to show?

7. In the cortical layer recovery section, Figure 3c showed that the accuracy of CeLEry to predict Layer 2 is only about 0.375 in Scenario 1 and 0.25 in Scenario 2, which seems too low for a classification task. Besides, the performance of the method didn't improve significantly after applying the data augmentation step, and the accuracy even decreased in Scenario 4. The authors should discuss these results.

8. The authors also applied CeLEry on two different species datasets to show the method worked well. However, I'm not sure this attempt is meaningful enough. Could the authors discuss that are there any real application scenarios that reconstruct the spatial location information of mouse scRNA-seq data using a human spatial transcriptomic data reference?

9. The authors applied CeLEry on two brain scRNA-seq datasets and evaluated the performance of spatial domain reconstruction of single cells. I'm wonder how CeLEry performed on other tissue datasets with more complex structure such as tumors. As mentioned above, I'm also curious about the performance of recovering cell locations of single cells.

10. In Figure 6e, I don't know if I misunderstood it, the maximum and minimum values of proportion differed only by about 0.1, does this indicate that the results of the method are not very good? In my opinion, the differences between spatial domains should be distinct enough.

11. In Line 260, the authors mentioned that they first enhanced the gene expression resolution of the ST training data using TESLA. I'm curious about how much the performance of CeLEry improved after applying this strategy. Could the authors provide the results of CeLEry without enhancing the gene expression resolution of the ST training data for comparison?

12. In the AD application, I find the authors still utilize the human DLPFC data from normal brain tissues to train the model. I don't believe this is the best approach because there would be major changes in the process of AD, and these changes would influence the spatial variation and heterogeneity of cells.

13. The authors provide code and vignettes on Github, but it's unclear if they have provided code to reproduce all results in the manuscript. It would be great if the authors could also provide the analysis code on Github for reproducibility.

Minor:

1. In Figure 3, 4, 5, and 6, the colors chosen for layers, probability, and proportion need to be more contrasting as it is currently challenging to distinguish between them in the figures.

2. When applying multiple training data, is the input or processing process of this method the same as single training data? I don't find the instructions in the "method" section, and the authors should explain them in detail.

3. Line 492, what are n and p represent?

4. Line 498, what are the parameters for gene clustering?

5. Line 554, the authors mentioned that they use ReLU as the activation, but I noticed that they chose the Sigmoid in the last layer of the DNN model. Perhaps the authors could elaborate on it.

Reviewer #2 (Remarks to the Author):

In this manuscript, the authors presented a new method, CeLEry, to recover cell locations in single-cell RNA-seq data. It is very exciting to see another new method to tackle this challenge. The method is novel, the topic is interesting, and the overall writing is clear. While the authors performed evaluations using several datasets, all of these datasets are about brains (including mouse and human). The natural question is, is this method applicable for brain tissue only? Currently, many spatial transcriptomics data are indeed sequenced on brain tissue, but there is also wide interest in other tissues such as the heart, lung, liver, pancreas, etc. However, this new method seems to be a brain-specific model. It would be more appropriate to add "on brains" in the title so that users can clearly know its applicable tissue type. Otherwise, the authors might want to demonstrate the method's good performance on other tissue types as well.

Here are other comments.

1. In the Introduction section, the following sentence needs to be further explained. "However, due to the lack of consideration of location dependency, the predicted locations for a given cell can be far apart from each other, making interpretation of the results difficult."
2. On page 5, the authors claimed, "To computationally enlarge the sample size of the training data, CeLEry has a data augmentation procedure to generate replicates of the original ST data". However, the computationally generated replicates can be very different from the actual experimentally generated replicates in various aspects, including gene expression variation, cell abundance distribution, and cell type composition. these replicates could contain a large donor-to-donor variation. Even if the replicates are from the same donor, they could be very different. Such variation should be confounded with the individual-to-individual variation, which is often of great interest to studies. However, the author only uses one slide. For example, "learn the mean and standard deviation of the gene expression embedding distribution using a variational autoencoder". The mean and sd here are statistics within one slide, which differs from the variation across slides, but the authors use this one-slide information to generate multiple slides. The variation across slides has not been guaranteed. Authors should systematically benchmark how one slide can produce this multiple-slide variation in a way that is comparable to actual multiple experimental replicates.
3. Similarly, on page 6, " the augmentation-generated gene expression shows variations among replicates". Could authors use some experimentally generated replicates from the same tissue type to demonstrate the validity of these variations in these replicates augmented from only one slide?
4. In Fig 2A, the authors showed four examples from each gene cluster to demonstrate the cluster patterns. Example genes could only show a very small part of the whole cluster. Could the authors show the averaged pattern for each of these clusters and also visualize the standard deviation on the slide?
5. In Fig 3C, the performance of CeLEry with augmentation was provided. Can the authors provide the performance when augmenting to n slides and discuss the performance with different values of n?
6. Authors described many accuracies on page 7. For example, the authors wrote, " The top-1 and top-2 accuracies for CeLEry are 53.8% and 89.2%, respectively, for Scenario 1, and 44.0% and 82.6%, respectively, for Scenario 2. However, there is no figure for these statistics. A quick guess would be Fig 3c since these descriptions were shown right after the introduction of Fig 3c. However, Fig 3c is about the accuracy for each layer, which is different from the definition of top-1 and top-2 accuracies. Could the authors add a figure about this? Also, the information in Fig 3c seems interesting but has not been well discussed. What is the "Exact + Neighbour"?
7. On page 8, the model was trained using human brain data and applied to mouse brain data. What will the performance be if the training and testing data are switched? Also, can the evaluation be performed with data augmentation (in both directions)?
8. On page 9, the authors wrote, " it is unclear how it performs when the test data are generated from scRNA-seq", therefore, they "predict the layer information for cells in the snRNA-seq data". scRNA-seq and snRNA-seq are different sequencing technologies. It is quite misleading when these two are

mixed. Which one is actually “the main task that we are interested in?”

9. On page 9, “We included 1,697 genes that were selected from the joint set of the top 200 enriched genes for each of the six layers in each of the four tissue sections in the analyses.” If the spatial location information (layers here) has been passed to CeLEry through these genes, which are layers’ markers, then has the answer (spatial location) already been told to CeLEry?

10. On page 10, “As a comparison, we also predicted the layer information for cells in the Hodge et al. data using Tangram.” This comparison does not seem fair since, as mentioned in the above comment, CeLEry has the layer-specific gene information passed onto the model, which Tangram does not have, and therefore CeLEry seems to have a benefit in this comparison. Do other analyses/evaluations also have this issue?

11. On page 11, “The left part of the brain was used to collect the snRNA-seq data, where the cells were collected from four different regions and sequenced in three snRNA-seq datasets as shown in Fig. 6B.” In Fig 6B, only three arrows from regions ACD, how about the region B?

12. On page 12, the authors wrote, “we first enhanced the gene expression resolution of the ST training data using TESLA”. Since this enhancement was not conducted in other analyses/evaluations in this manuscript, could the author include a comparison between using and not using enhancement? Users will benefit from it when applying CeLEry to their datasets. It would also be helpful if authors could comment on the scenarios where users are recommended (or not) to use enhancement.

13. On page 12, since the performance of TESLA could be a confounding factor in the ultimate performance of both CeLEry and Tangram, could authors include at least two enhancement methods (including TESLA) to demonstrate the superiority of CeLEry is not affected by the enhancement method?

14. On page 13, the authors wrote, “the reconstructed 2D gene expression maps based on CeLEry’s recovered 2D locations are more similar to the ground truth than those of Tangram (Fig. 7C).” However, in Fig 7C, it seems that the outputs from CeLEry and Tangram are quite similar visually. It is hard to tell which one is more similar to the ground truth. Can authors provide a quantification of this similarity to the ground truth?

POINT-TO-POINT RESPONSE TO REVIEWERS' COMMENTS

We sincerely thank the reviewers for their constructive comments, which have helped improve the exposition of our manuscript. We have made substantial improvements to our manuscript and highlighted the revised/new parts in red for ease of reviewing. Below are our point-by-point responses to the reviewers' comments. The original reviewers' comments are in bold italics and our responses are in normal font colored in blue.

Reviewer #1 (Remarks to the Author):

In this manuscript, Zhang et al developed a computational method, CeLEry, to recover the location information for cells in scRNA-seq data by utilizing the gene expression and spatial location relationships learned from spatial transcriptomic data. They also proposed a data augmentation method that utilizes a VAE model to generate realistic replicates of the original spatial transcriptomic data, to increase the sample size of the training data to learn more relationships between the gene expression and spatial location in spatial transcriptomic data. They mainly compared the performance of CeLEry with Tangram on both spatial transcriptomic data and scRNA-seq data, and finally, they applied CeLEry to an AD snRNA-seq data to recover cortical layers information of cells.

Since a large amount of scRNA-seq data have been generated, reconstructing spatial location information of cells is meaningful. However, I think there is still some further validation and expansion needed to make the significance of the CeLEry clearer. My main concerns are listed below, hoping these are useful to the authors.

We thank the reviewer for their encouraging comments and constructive critiques, which we address below.

Major:

1. One key concern of mine here relates to the theoretical justification of this method. The method trains the model on spatial transcriptomic data and then utilizes the trained model for scRNA-seq data. However, these (spatial transcriptomic data and scRNA-seq data) represent two different data types. The spatial transcriptomic data used in the manuscript is 10x Visium, it doesn't reach the single-cell resolution and each spot contains cells with different cell types. It seems a bit "forced" to reconstruct spatial location information of single cells using the mixed spatial transcriptomic data as a reference. Some validation experiments should be performed (perhaps comparing the spatial reconstruction results of scRNA-seq data of this method with deconvolution results of spatial transcriptomic data reference is a way to try).

We thank the reviewer for this thoughtful comment and regret that our description of CeLEry was unclear. The method that we proposed is generic and can take any type of spatial transcriptomics data as input to train the location recovery model. We agree with the reviewer that the use of 10x Visium data as a reference is not ideal because Visium doesn't have a single-cell resolution. As such, we have analyzed multiple datasets that have single-cell resolution. Below we describe the details of these additional analyses.

First, we analyzed a mouse brain dataset generated using MERSCOPE (commercialized version of MERFISH, data downloaded from <https://info.vizgen.com/mouse-brain-data>), which has a single-cell resolution. In this benchmark evaluation, we split the entire dataset into half (left brain) and half (right brain). We trained the CeLEry model with the left brain and tested it on the right brain. We also trained the CeLEry model with the right brain and then tested it on the left brain. This analysis allows us to recover cell locations for the entire brain. Since the spatial location information for each cell is known, this experiment allows us to evaluate the location recovery accuracy. As shown in revised Fig. 6 (also shown below), CeLEry outperformed Tangram, spaOTsc, and novoSpaRc in a) reconstructing the overall spatial relationship of cells; b) and c) retaining pairwise distance between cells; d) predicting cell locations with relatively uniform error rates across the tissue; and e) reconstructing spatial gene expression patterns.

Revised Fig. 6: 2D location recovery for single cells in MERSCOPE mouse brain data. **a**, The true cell location maps and predicted cell location maps generated by CeLEry, Tangram, spaOTsc, and novoSpaRc, with cells annotated according to cluster assignment. The left half of cell maps were obtained by predicting the cells from the left brain using models trained using cells from the right brain. The predicted results were then flipped to the left brain and scaled to match the size of the left brain. The same procedure was applied when processing the cells from the right brain. **b**, Scatter density plot comparing the true pairwise spatial distance versus the predicted pairwise spatial distance between a pair of cells. Color in the plot indicates cell pair density. **c**, Barplot of the Pearson correlation between the true pairwise distance and the predicted pairwise distance for all cell pairs. **d**, Euclidean distance between the true and predicted locations for each cell in the

test data. **e**, Recovered gene expression map of two randomly selected genes *ADRB1* and *GRM5*, based on the predicted locations in **a**, with color indicating relative gene expression.

Next, we analyzed a single-cell ST dataset generated from a liver cancer patient using the MERSCOPE platform (data downloaded from <https://info.vizgen.com/merscope-ffpe-solution>). In this analysis, we split the data into training (50%) and testing (50%) and evaluated the accuracy of the recovered cell locations. As shown in revised **Fig. 7**, CeLEry outperformed Tangram, spaOTsc, and novoSpaRc in a) reconstructing the overall spatial relationship of cells; b) and c) retaining pairwise distance between cells; d) predicting cell locations with relatively uniform error rates across the tissue; and e) reconstructing spatial gene expression patterns.

Revised Fig. 7: 2D location recovery for single cells in MERSCOPE human liver cancer data. **a**, The true cell maps and predicted cell maps generated by CeLEry, Tangram, spaOTsc, and novoSpaRc, with cells annotated according to cluster assignment for the test cells. **b**, Scatter density plot comparing the true pairwise spatial distance versus the predicted pairwise spatial distance. Color in the plot indicates cell pair density. **c**, Barplot of the Pearson correlation between the true pairwise distance and the predicted pairwise distance for all cell pairs. **d**, Euclidean

distance between the true and predicted locations for each cell of the test data. **e**, Recovered gene expression map of two randomly selected genes *CDKN1B* and *TGFB1*, based on the predicted locations in **a**, with color indicating relative gene expression.

Finally, we analyzed a single-cell ST dataset generated from a breast cancer patient using the 10x Genomics recently released Xenium platform. Again, we randomly split the data into training (50%) and testing (50%) and evaluated the performance of CeLEry following similar strategies for the liver cancer data. As shown in revised **Fig. 8**, CeLEry outperformed Tangram, spaOTsc, and novoSpaRc in a) reconstructing the overall spatial relationship of cells; b) and c) retaining pairwise distance between cells; d) predicting cell locations with relatively uniform error rates across the tissue; and e) reconstructing spatial gene expression patterns.

Revised Fig. 8: 2D location recovery for single cells in 10X Xenium breast cancer data. a, The true cell location maps and the predicted cell location maps generated by CeLEry, Tangram, spaOTsc, and novoSpaRc, with cells annotated according to cluster assignment for the test cells. **b**, Scatter density plot comparing the true pairwise spatial distance versus the predicted pairwise spatial distance. Color in the plot indicates cell pair density. **c**, Barplot of the Pearson correlation between the true pairwise distance and the predicted pairwise distance for all cell pairs. **d**,

Euclidean distance between the true and predicted locations for each cell of the test data. **e**, Recovered gene expression map of two randomly selected genes GATA3 and CEACAM6, based on the predicted locations in **a**, with color indicating relative gene expression.

Through the above single-cell resolution ST based benchmark evaluations, we have shown that CeLEry can utilize single-cell resolution ST data as reference. We hope these results demonstrate CeLEry's ability in utilizing diverse ST data in building its cell location prediction model.

2. All the validation and applications in the manuscript are based on the same set of spatial transcriptomic data (DLPFC). I would have to see results showing a good spatial reconstruction of different tissues to determine that this method is well scalable.

We appreciate the reviewer's constructive suggestion. As shown in our response to your previous comment, we have added three benchmark evaluations that didn't use the DLPFC data as the spatial reference. We hope these new results demonstrate CeLEry's scalability in analyzing datasets generated from diverse ST platforms in different tissues.

3. Although the idea of this manuscript is to recover cell locations of single cells, the authors mainly performed experiments that recover the spatial domain of cells. It's more like a classification task. In my opinion, recovering cell locations of single cells is more meaningful, but the authors only did a validation experiment of this in mouse brain data. I would like to see more results from the reconstruction of cell locations. The authors could create two long vectors (one for the reconstruction result and one for the ground truth) containing the distance between every pair of cells and then calculate the correlation between them to evaluate the performance of their method.

We appreciate the reviewer's comment and constructive suggestion on how to evaluate the performance of the method. To demonstrate CeLEry's ability in recovering 2D cell locations, we have performed additional evaluations in this revision. Please see our response to your comment #1.

While spatial domain recovery is not as interesting and challenging as 2D cell location recovery, in the revised manuscript, we have shown that it can be adapted to predict cell types, and for this simpler task, CeLEry still outperformed the other competing methods.

Supplementary Fig. 11. Cell type recovery for 10x Xenium breast cancer data. **a**, The true cell types of cells in the breast cancer data annotated by 10x Genomics. **b**, The predicted cell types

visualization generated by CeLEry, Tangram, SpaOTsc and novoSpaRc, whose overall prediction accuracies are 96.3%, 88.4%, 85.1% and 86.3%, respectively.

In the revised Discussion, we added a paragraph about this new results and also clarified the major difference between CeLEry and other methods. The text is pasted below:

*“We note that most of the existing methods focus on ST-centric analysis. Since their goal is to utilize scRNA-seq data to help improve the quality of ST, recovering the exact 2D-location with high accuracy for each scRNA-seq cell is not their primary purpose. Indeed, the main purpose of these ST-centric methods is to map a scRNA-seq cell to a specific tissue location that has the same cell type. Since each cell type often includes many cells, as long as a cell from that type is mapped to the corresponding location, it is considered a correct mapping. Methodologically, this is a much easier task than requiring a scRNA-seq cell to be mapped to the correct tissue location. Although CeLEry was developed for the purpose of recovering cell locations, its framework is sufficiently general to be adapted to map cell types. To illustrate this functionality, we trained the spatial domain prediction model by treating each cell type as a spatial domain. This adaptation is reasonable as cell types do not possess an ordinal relationship. We conducted additional analysis on the 10x Xenium breast cancer data and found that when evaluating the performance of CeLEry by cell type mapping, the accuracy is substantially improved to 0.96 (**Supplementary Fig. 11**) and outperformed Tangram, spaOTsc and novoSpaRc. This result indicates that CeLEry is versatile in its usage.”*

4. I appreciate that the authors benchmark their method, but would argue that their choice of method (Tangram) to benchmark against is a bit surprising. From my perspective, I think CeLEry is more like a method for spatial reconstruction of single cells. To make the comparison more relevant I would encourage the authors to benchmark their method with other spatial reconstruction methods, for example, novoSpaRc, SpaOTsc, etc.

We thank the reviewer for this suggestion. In this revision, we have added comparisons with novoSpaRc and spaOTsc for all benchmark datasets analyzed in the paper. These include the following datasets:

- 1) Benchmark evaluation for cortical layer recovery in human dorsolateral prefrontal cortex;
- 2) 2D location recovery in mouse brain with 10x Visium data as spatial reference;
- 3) Benchmark evaluation for 2D location recovery in mouse brain with MERSCOPE data as spatial reference;
- 4) Benchmark evaluation for 2D location recovery in liver cancer with MERSCOPE data as spatial reference;
- 5) Benchmark evaluation for 2D location recovery in breast cancer with 10x Xenium data as spatial reference.

In all evaluations, CeLEry outperformed Tangram, spaOTsc, and novoSpaRc.

Through these evaluations, we have found that while spaOTsc performed reasonably well in some data, this method cannot handle large datasets. Indeed, when analyzing the liver cancer MERSCOPE data, spaOTsc can only analyze 25% of the cells, which will limit its applications in large-scale single-cell studies. Tangram and novoSpaRc also have scalability issues in our evaluations. We have added this observation in the revised Results (text pasted below):

“The performance of different methods was evaluated by comparing the recovered cell maps to the original maps, as presented in Fig. 7a. It is noteworthy that Tangram, SpaOTsc, and novoSpaRc are unable to process the testing data in its entirety, as the sample size of the test data exceeds their computational capabilities. Specifically, Tangram, SpaOTsc, and novoSpaRc were only able to process 50%, 25%, and 30% of the test data, respectively. For the sake of comparison, a subset of 25% of the test cells was presented in Fig.7a.”

5. In the data augmentation section, if I understand correctly, the authors created 2D gene expression

graphs (like “photos”) based on the spatial location and gene expression of spatial transcriptomic data and utilized a CNN network to extract the low dimensional feature representations. However, for the image-based spatial transcriptomic data like slide-seq, STARmap, or MERFISH, due to their non-Euclidean structure spatial locations, I don’t know if this method is still work? If not, the limitations should be addressed in the discussion.

Thank you for this insightful comment. You are correct that the data augmentation procedure would not work for ST data that do not have a grid layout. We have added this limitation in the Discussion of the revised manuscript (text pasted below),

“Our data augmentation procedure only works for ST data that have a grid layout, e.g., 10x Visium. ST data with non-grid layout, e.g., SLIDE-seq, MERFISH, or 10x Xenium, would not benefit from this procedure. However, since the number of cells in MERFISH or Xenium data is typically large, data augmentation may be unnecessary.”

6. In Figure 2b, the augmentation results of CALB1 don’t seem very good. Does the use of these “realistic replicates” introduce additional noise? Besides, is there any particular reason for only selecting these two genes rather than the six gene clusters to show?

We regret that we only presented two randomly selected genes in the original submission. In this revision, we have included more genes to demonstrate CeLEry can generate “realistic replicates” through data augmentation. In **Supplementary Fig. 2**, we present the performance of data augmentation for a randomly selected gene for each of the six gene clusters.

7. In the cortical layer recovery section, Figure 3c showed that the accuracy of CeLEry to predict Layer 2 is only about 0.375 in Scenario 1 and 0.25 in Scenario 2, which seems too low for a classification task. Besides, the performance of the method didn’t improve significantly after applying the data augmentation step, and the accuracy even decreased in Scenario 4. The authors should discuss these results.

We thank the reviewer for this thoughtful comment. Admittedly, the prediction of spots on Layer 2 is challenging because this layer is very thin and thus the number of spots in the training data is limited. Nevertheless, CeLEry outperformed Tangram, spaOTsc and novoSpaRc, particularly in top-2 accuracies. Moreover, we observed improved performance after increasing the training sample size of Layer 2 either through data augmentation (CeLEry (aug) in Scenario 2) or through including multiple biological replicates (Scenario 4) (revised **Fig. 3e**, also pasted below).

Revised Fig. 3e

The data augmentation procedure can serve as a surrogate for enlarging the sample size. However, it cannot replace biological replicates completely. Together with Reviewer 2's comments on biological replicates, we have now discussed this in the data augmentation section (text pasted below).

“To determine the similarity between the generated replicates and the actual biological replicates, we also analyzed a 10x Visium dataset generated by Maynard et al. ¹⁹ (denoted as the LIBD data). The LIBD data were generated from three postmortem brains with each brain having four tissue sections obtained from the dorsolateral prefrontal cortex (DLPFC). This dataset allows us to examine whether the generated replicates from our data augmentation procedure resemble those actual biological replicates. For illustration purposes, we randomly selected three genes CAMK2N1, TMSB10, and HPCA, and visualized their expression (Supplementary Fig. 3). While the artificial replicates were smoother than the biological replicates, they displayed similar patterns of gene expression variation across replicates as seen in the biological replicates. These results indicate that while artificial replicates cannot completely replace biological replicates, they can serve as a substitute when biological replicates are not available.”

8. The authors also applied CeLEry on two different species datasets to show the method worked well. However, I'm not sure this attempt is meaningful enough. Could the authors discuss that are there any real application scenarios that reconstruct the spatial location information of mouse scRNA-seq data using a human spatial transcriptomic data reference?

We agree with the reviewer that this example is not very meaningful. As such, we have decided to delete this example in the revised manuscript.

9. The authors applied CeLEry on two brain scRNA-seq datasets and evaluated the performance of

spatial domain reconstruction of single cells. I'm wonder how CeLEry performed on other tissue datasets with more complex structure such as tumors. As mentioned above, I'm also curious about the performance of recovering cell locations of single cells.

Thanks for this insightful comment. Following your suggestion, we performed benchmark evaluations using two cancer datasets: 1) a liver cancer dataset generated using MERSCOPE (results shown in revised **Fig. 7**), and 2) a breast cancer dataset generated using 10x Xenium (results shown in revised **Fig. 8**). For both datasets, CeLEry outperformed Tangram, spaOTsc, and novoSpaRc.

10. In Figure 6e, I don't know if I misunderstood it, the maximum and minimum values of proportion differed only by about 0.1, does this indicate that the results of the method are not very good? In my opinion, the differences between spatial domains should be distinct enough.

We agree with the reviewer that this example is not very convincing. Therefore, we have decided to delete this example in the revised manuscript.

11. In Line 260, the authors mentioned that they first enhanced the gene expression resolution of the ST training data using TESLA. I'm curious about how much the performance of CeLEry improved after applying this strategy. Could the authors provide the results of CeLEry without enhancing the gene expression resolution of the ST training data for comparison?

The main reason to enhance gene expression resolution using TESLA is that the original Visium data do not have a single-cell resolution. An ideal training data is a ST dataset that covers the entire transcriptome and has a single-cell resolution. Since such data are not available, we used TESLA to enhance the gene expression resolution in the training data. Following your suggestion, we have included results before TESLA enhancement in **Supplementary Fig. 4b**. Not surprisingly, the performance of all methods became worse, however, CeLEry still outperformed the other methods. We have added the description of these results in the Results section of the revised manuscript (text pasted below),

*"In the above analyses, we used enhanced gene expression data obtained from TESLA as the spatial reference. The reason for doing this is that the original Visium data do not have a single-cell resolution. The ideal spatial reference is a ST dataset that covers the entire transcriptome and has a single-cell resolution. Since such data are not available, we used TESLA to enhance the spatial resolution of Visium. While this does not achieve a single-cell resolution, it offers a practical solution when ST data with the single-cell resolution are not available. As shown in **Supplementary Fig. 4b**, when using the original Visium data as the spatial reference, the performance of all methods became worse, however, CeLEry still outperformed the other methods. In general, when single-cell resolution ST training data are not available, we would recommend that users enhance the gene expression resolution first."*

12. In the AD application, I find the authors still utilize the human DLPFC data from normal brain tissues to train the model. I don't believe this is the best approach because there would be major changes in the process of AD, and these changes would influence the spatial variation and heterogeneity of cells.

Thank you for this insightful comment. To address your concern, we reanalyzed the data using a revised analysis strategy. In this revised analysis, we removed those genes that are differentially expressed between the AD and the control groups in the snRNA-seq reference by a Wilcoxon rank sum test. Specifically, a gene is removed if the log fold change is greater than 1 and the adjusted p-value (adjusted by Benjamin-Hochberg) is less than 0.05. Then, we used the remaining genes to predict the layers of single cells. This revised analysis can avoid potential confounding caused by disease. As shown in revised **Fig. 4**, the new analysis produced results that are generally consistent with the earlier findings except that fewer cells were predicted to Layer 4 (revised **Fig. 4b**). Similar to our previous findings, the predictions on the inner layers had higher uncertainty than those of the outer layers (revised **Fig. 4c**), whereas the differences of prediction uncertainty across different cell types were smaller (revised **Fig. 4d**). However, the varying compositions of cell types in each layer (revised **Fig. 4e**) identified in the new analysis were slightly different, with L1 being dominated by astrocytes, whereas L2-L5 had

more neuronal cells. The oligodendrocytes were mainly mapped to L6 and especially the white matter (revised Fig. 4e). Comparing subjects of different disease statuses (revised Fig. 4f), we observed the loss of neuronal cells in Layers 5 and 6, and as the disease progressed the loss of neuronal cells in Layer 4 was also observed. We have updated the Results section in the revised manuscript.

13. The authors provide code and vignettes on Github, but it's unclear if they have provided code to reproduce all results in the manuscript. It would be great if the authors could also provide the analysis code on Github for reproducibility.

Following your suggestion, we have included all Python and R codes for data analyses and figure generation on Github (https://github.com/QihuangZhang/CeLEry/tree/main/code_paper).

Minor:

1. In Figure 3, 4, 5, and 6, the colors chosen for layers, probability, and proportion need to be more contrasting as it is currently challenging to distinguish between them in the figures.

Following your suggestion, we have revised Figure 3 by adopting color schemes with higher contrast. The original Figures 4-6 were removed in this revision. In the new Figures, we used more contrasting colors.

2. When applying multiple training data, is the input or processing process of this method the same as single training data? I don't find the instructions in the "method" section, and the authors should explain them in detail.

We regret that our original description was unclear. We performed analysis using multiple training data only for the layer prediction task. In this analysis, we first performed gene expression normalization and logarithm transformation for each training data separately. Then, we subset the common genes shared by all datasets and merged the gene expression and layer information to generate a combined training dataset. We have added these detailed descriptions in the revised Methods section (text pasted below).

"For the analysis of cortical layers in the LIBD data that involve multiple training tissue sections, we first perform gene expression normalization and logarithm transformation for each training section separately. Then, we subset the common genes shared by all sections and merge the gene expression and layer information to generate a combined training dataset."

3. Line 492, what are n and p represent?

Here, n represents the number of spots and p represents the number of genes. We have revised the text in the Methods section accordingly (pasted below).

"Let $Z \in R^{n \times p}$ be the gene expression matrix obtained from a ST dataset in which rows represent spots and columns represent genes, n is the number of spots, and p is the number of genes."

4. Line 498, what are the parameters for gene clustering?

There is only one hyperparameter in K-means clustering, which is the number of clusters. In the analyses, the number of clusters was set to 50 by default.

5. Line 554, the authors mentioned that they use ReLU as the activation, but I noticed that they chose the Sigmoid in the last layer of the DNN model. Perhaps the authors could elaborate on it.

We regret that the original description was not clear. We employed ReLU as an activation function for the hidden layers, but used the Sigmoid for the last layer (output layer). The reason to use Sigmoid is to scale the output into the $[0, 1]$ range for both the 2D-location recovery and the layer (or spatial domain) recovery tasks. Rescaling 2-D coordinates into the $[0, 1]$ range will facilitate the comparison of results across different datasets. For layer

(or spatial domain) recovery, the $[0, 1]$ -ranged output represents the probability of assigning an observation to a layer (or spatial domain). We have now clarified the use of activation functions in the revised Methods (text pasted below).

“We use ReLU as the activation function for the hidden layers. For the output layer, we use Sigmoid as the activation function in order to scale the output into the $[0, 1]$ range. Rescaling 2-D coordinates into the $[0, 1]$ range will facilitate the comparison of results across different datasets. For layer (or spatial domain) recovery, the $[0, 1]$ -ranged output represents the probability of assigning an observation to a layer (or spatial domain).”

Reviewer #2 (Remarks to the Author):

In this manuscript, the authors presented a new method, CeLEry, to recover cell locations in single-cell RNA-seq data. It is very exciting to see another new method to tackle this challenge. The method is novel, the topic is interesting, and the overall writing is clear. While the authors performed evaluations using several datasets, all of these datasets are about brains (including mouse and human). The natural question is, is this method applicable for brain tissue only? Currently, many spatial transcriptomics data are indeed sequenced on brain tissue, but there is also wide interest in other tissues such as the heart, lung, liver, pancreas, etc. However, this new method seems to be a brain-specific model. It would be more appropriate to add “on brains” in the title so that users can clearly know its applicable tissue type. Otherwise, the authors might want to demonstrate the method’s good performance on other tissue types as well.

We thank the Reviewer for their positive appraisal and thoughtful comments. While CeLEry was mainly motivated by problems that we encountered when analyzing brain data, the methods are generic. In this revision, we have added two cancer examples (revised **Fig. 7** for liver cancer, and revised **Fig. 8** for breast cancer), where we showed CeLEry’s promising performance in recovering cell locations for non-brain tissues.

Here are other comments.

1. In the Introduction section, the following sentence needs to be further explained. “However, due to the lack of consideration of location dependency, the predicted locations for a given cell can be far apart from each other, making interpretation of the results difficult.”

We apologize for the lack of clarity. Tangram treats spots or cells in the spatial reference data as independent and hence the physical proximity of the spots/cells is not modeled. In contrast, CeLEry treats the spot/cell location information as a continuous variable, which can naturally account for location dependency when predicting a cell’s location. We have revised the Introduction in the revision (text pasted below).

“Tangram, spaOTsc, and NovoSpaRc all map the scRNA-seq cells to multiple locations in a probabilistic fashion. However, these methods treat locations as discrete variables, and the lack of consideration of the physical proximity of nearby locations can make the predicted locations for a given cell to be far apart from each other, which leads to difficulties in interpretation.”

2. On page 5, the authors claimed, “To computationally enlarge the sample size of the training data, CeLEry has a data augmentation procedure to generate replicates of the original ST data”. However, the computationally generated replicates can be very different from the actual experimentally generated replicates in various aspects, including gene expression variation, cell abundance distribution, and cell type composition. these replicates could contain a large donor-to-donor variation. Even if the replicates are from the same donor, they could be very different. Such variation should be confounded with the individual-to-individual variation, which is often of great interest to studies. However, the author only uses one slide. For example, “learn the mean and standard deviation of the gene expression embedding distribution using a variational autoencoder”. The mean and sd here are statistics within one slide, which differs from the variation across slides, but the authors use this one-slide information to generate multiple slides. The variation across slides has not been guaranteed. Authors should systematically benchmark how one slide can produce this multiple-slide variation in a way that is comparable to actual multiple experimental replicates.

Thank you for these thoughtful comments. Our data augmentation procedure currently only introduces variations by learning from the gene expression patterns in one slide. We agree that there exist slide-to-slide and individual-to-individual variations. Ideally, such variations should be considered. However, this is a challenging problem because such modeling requires multiple reference datasets to be mapped to the same 2D-coordinate space. Such mapping is not always possible; for example, for biopsies obtained from different tumor samples collected from different donors, it is not obvious how such data can be mapped to the same 2D-coordinate space. Our main motivation to do data augmentation is to “enlarge” the training sample size when only one training slide is

available. Currently, all of the methods, including CeLEry, can only take one slide as a training sample when the goal is to recover the 2D locations of cells. (Note: CeLEry can be utilized multiple training slides only for the spatial domain/layer recovery task because this does not require the multiple tissue sections to be mapped to the same 2D-coordinate space). We are currently extending CeLEry to jointly model multiple slides without the requirement of mapping all slides into the same 2D-coordinate space, but we believe that implementing and testing such an approach extends beyond the scope of the current study. Nevertheless, it is an exciting potential future direction.

To address your concerns that the variations generated by our data augmentation procedure may not reflect the true biological variations seen in multiple tissue slides, we conducted additional analyses. Specifically, we analyzed the LIBD data, which includes 12 tissue sections generated from 3 donors (4 sections per donor). This dataset allows us to evaluate whether the variations generated from our data augmentation procedure resemble those seen in real biological replicates. **Supplementary Fig. 3** (also shown below) shows the expression pattern for the artificial replicates (based on data augmentation from 1 tissue section) and the 3 biological replicates obtained from the same brain.

Supplementary Fig. 3: Gene expression map comparison between the biological replicates (tissue ID 151674, 151675, and 151676) and the artificial replicates produced by data augmentation procedure. We randomly selected three genes (a) CAMK2N1, (b) TNSB19, and (c) HPCA from the LIBD human DLPFC data as examples.

We observed that the artificial replicates generally show smoother gene expression patterns than the biological replicates. Although not completely reflecting the true biological variations, the artificial replicates did display a similar gene expression variation across replicates as seen in the biological replicates. These results indicate that while artificial replicates cannot completely replace biological replicates, they can serve as a substitute when biological replicates are not available. We have added these new results in the Results section of the revised manuscript (text also pasted below).

“To determine the similarity between the generated replicates and the actual biological replicates, we also analyzed a 10x Visium dataset generated by Maynard et al.¹⁹ (denoted as the LIBD data).

The LIBD data were generated from three postmortem brains with each brain having four tissue sections obtained from the dorsolateral prefrontal cortex (DLPFC). This dataset allows us to examine whether the generated replicates from our data augmentation procedure resemble those actual biological replicates. For illustration purposes, we randomly selected three genes CAMK2N1, TMSB10, and HPCA, and visualized their expression (**Supplementary Fig. 3**). While the artificial replicates were smoother than the biological replicates, they displayed a similar pattern of gene expression variation across replicates as seen in the biological replicates. These results indicate that while artificial replicates cannot completely replace biological replicates, they can serve as a substitute when biological replicates are not available.

3. Similarly, on page 6, “the augmentation-generated gene expression shows variations among replicates”. Could authors use some experimentally generated replicates from the same tissue type to demonstrate the validity of these variations in these replicates augmented from only one slide?

We believe our response to your previous comment #2 answered this question.

4. In Fig 2A, the authors showed four examples from each gene cluster to demonstrate the cluster patterns. Example genes could only show a very small part of the whole cluster. Could the authors show the averaged pattern for each of these clusters and also visualize the standard deviation on the slide?

Thanks for this suggestion. In this revision, we have included example genes for all clusters in **Fig. 2a** (also pasted below).

Revised Fig. 2a

Following your suggestion, we also visualized the average pattern and the spot-wise standard deviation of the gene expression across all genes in a cluster in **Supplementary Fig. 1** (also pasted below).

Supplementary Fig. 1. Spotwise mean and standard error of gene expressions for the six randomly selected gene clusters obtained from the mouse posterior brain data.

5. In Fig 3C, the performance of CeLEry with augmentation was provided. Can the authors provide the performance when augmenting to n slides and discuss the performance with different values of n ?

We investigated the performance of CeLEry when augmenting to n slides and these results were shown in Fig. 3b. Choosing a larger n does not guarantee a higher accuracy performance. As n becomes larger more proportion of training data will come from artificial generation and hence the model gives smaller weight to the original data. In real data applications, we recommend the user to explore the best number of replicates by performing cross-validation.

6. Authors described many accuracies on page 7. For example, the authors wrote, “The top-1 and top-2 accuracies for CeLEry are 53.8% and 89.2%, respectively, for Scenario 1, and 44.0% and 82.6%, respectively, for Scenario 2. However, there is no figure for these statistics. A quick guess would be Fig 3c since these descriptions were shown right after the introduction of Fig 3c. However, Fig 3c is about the accuracy for each layer, which is different from the definition of top-1 and top-2 accuracies. Could the authors add a figure about this? Also, the information in Fig 3c seems interesting but has not been well discussed. What is the “Exact + Neighbour”?

We regret that the original labels for the accuracy metrics are not clear. Following your suggestion, we have added revised Fig. 3d, which shows the top-1 and top-2 accuracies for each method. We report layer-wise accuracy as complementary information to the description in the main text.

In addition, we added more descriptions about **Fig. 3c** in the revision (text pasted below).

*“**Fig. 3c** shows the layer-specific prediction accuracy. In general, we found it is relatively easy for CeLEry to recover spots that originate from the white matter, and augmentation helped improve the prediction accuracy for most layers. In contrast, Tangram showed consistently low accuracy for all layers; novoSpaRc predicted with higher accuracy in white matter (WM) but relatively low accuracy in other layers; and spaOTsc, however, predicted most of the spots to L3 or L5 regardless of their true layers, showing imbalanced performance among layers with high accuracies for these two layers only but low accuracies for others.”*

In the original **Fig. 3** legend, “Exact” refers to “top-1” accuracy, and “Exact+Neighbour” refers to “top-2” accuracy. To avoid confusion, we have used “top-1” and “top-2” in the revised **Fig. 3**.

The revised **Fig. 3** is shown below.

revised Fig. 3. Cortical layer recovery for spots in the LIBD human DLPFC data. a, Four scenarios considered in the evaluation. These scenarios vary in the number of tissue sections in the training data and the source of the test data, representing situations with different degrees of location recovery difficulty. **b,** The overall layer prediction accuracy for Scenario 2 with different numbers of replicates obtained from the data augmentation procedure. **c,** Layerwise prediction accuracies under different scenarios using CeLEry without data augmentation, CeLEry with data augmentation (2 replicates), novoSpaRc, spaOTsc, and Tangram. The results for Tangram are missing for Scenarios 3 and 4 because Tangram can only take one tissue section as the training data. **d,** Overall top-1 and top-2 prediction accuracies for CeLEry, CeLEry with data augmentation, Tangram, spaOTsc, and novoSpaRc, under different scenarios. It is noteworthy that Tangram, spaOTsc, and novoSpaRc are not applicable for Scenarios 3 and 4. **e,** Visualization of the probabilities of assigning each spot to different layers (shown as different rows) in using CeLEry without data augmentation (Scenario 2), CeLEry with data augmentation (Scenario 2, 2 replicates), CeLEry with multiple training samples (Scenario 4), novoSpaRc, spaOTsc, and Tangram (Scenario 2). The ground truth cortical layer structure for the test sample is shown on the right.

7. On page 8, the model was trained using human brain data and applied to mouse brain data. What will the performance be if the training and testing data are switched? Also, can the evaluation be performed with data augmentation (in both directions)?

Reviewer #1 raised a similar concern. Since it is not very meaningful to predict mouse data using human data as training, we have decided to delete this example in the revision.

8. On page 9, the authors wrote, “it is unclear how it performs when the test data are generated from scRNA-seq”, therefore, they “predict the layer information for cells in the snRNA-seq data”. scRNA-seq and snRNA-seq are different sequencing technologies. It is quite misleading when these two are mixed. Which one is actually “the main task that we are interested in?”

We regret that our description of the CeLEry model was unclear. Our prediction model can be utilized to recover the location information for both scRNA-seq and snRNA-seq data. We have clarified this point in the revised Methods.

9. On page 9, “We included 1,697 genes that were selected from the joint set of the top 200 enriched genes for each of the six layers in each of the four tissue sections in the analyses.” If the spatial location information (layers here) has been passed to CeLEry through these genes, which are layers’ markers, then has the answer (spatial location) already been told to CeLEry?

The layer-specific marker genes were selected from the spatial reference data, not the query scRNA-seq data. The layer information of the reference data is known and CeLEry learns from this information to model the relationship between gene expression and layer information. In our analyses, the same set of layer-specific marker genes was used as input for methods (CeLEry, Tangram, spaOTsc, and novoSpaRc). Since the layer information is unknown in the query data, the answers have not been told to CeLEry in its prediction.

10. On page 10, “As a comparison, we also predicted the layer information for cells in the Hodge et al. data using Tangram.” This comparison does not seem fair since, as mentioned in the above comment, CeLEry has the layer-specific gene information passed onto the model, which Tangram does not have, and therefore CeLEry seems to have a benefit in this comparison. Do other analyses/evaluations also have this issue?

The gene selection was performed prior to the data analysis and hence CeLEry and all the compared methods (including Tangram) used the same genes in the analyses.

In this revision, we have added several stronger examples. Therefore, we decided to delete the applications to the Hodge et al. data in the revision.

11. On page 11, “The left part of the brain was used to collect the snRNA-seq data, where the cells were collected from four different regions and sequenced in three snRNA-seq datasets as shown in Fig. 6B.” In Fig 6B, only three arrows from regions ACD, how about the region B?

Since this example is not very convincing, we have decided to delete this data application in the revision.

12. On page 12, the authors wrote, “we first enhanced the gene expression resolution of the ST training data using TESLA”. Since this enhancement was not conducted in other analyses/evaluations in this manuscript, could the author include a comparison between using and not using enhancement? Users will benefit from it when applying CeLEry to their datasets. It would also be helpful if authors could comment on the scenarios where users are recommended (or not) to use enhancement.

Reviewer #1 raised a similar concern (their comment #11).

The main reason to enhance gene expression resolution using TESLA is that the original Visium data do not have a single-cell resolution. An ideal training data is a ST dataset that covers the entire transcriptome and has a single-cell resolution. Since such data are not available, we used TESLA to enhance the gene expression resolution in the training data. Following your suggestion, we have included results before TESLA enhancement in **Supplementary Fig. 4b**. Not surprisingly, the performance of all methods became worse, however, CeLEry still outperformed the other methods. We have added the description of these results in the Results section of the revised manuscript (text pasted below),

*“In the above analyses, we used enhanced gene expression data obtained from TESLA as the spatial reference. The reason for doing this is that the original Visium data do not have a single-cell resolution. The ideal spatial reference is a ST dataset that covers the entire transcriptome and has a single-cell resolution. Since such data are not available, we used TESLA to enhance the spatial resolution of Visium. While this does not achieve a single-cell resolution, it offers a practical solution when ST data with the single-cell resolution are not available. As shown in **Supplementary Fig. 4b**, when using the original Visium data as the spatial reference, the performance of all methods became worse, however, CeLEry still outperformed the other methods. In general, when single-cell resolution ST training data are not available, we would recommend that users enhance the gene expression resolution first.”*

13. On page 12, since the performance of TESLA could be a confounding factor in the ultimate performance of both CeLEry and Trangran, could authors include at least two enhancement methods (including TESLA) to demonstrate the superiority of CeLEry is not affected by the enhancement method?

Thank you for this insightful comment. Gene expression resolution enhancement is a challenging task. To the best of our knowledge, besides TESLA, BayesSpace and XFuse are the only two methods that could enhance gene expression resolution in spatial transcriptomics data. BayesSpace enhances the spatial resolution by splitting the total gene expression in a spot into expression values for 6 sub-spots, but BayesSpace does not provide specific location information for the sub-spots, which makes it difficult to utilize BayesSpace enhanced gene expression as input for cell location recovery. XFuse can enhance gene expression resolution at the pixel level, however, XFuse is extremely slow -- training a single Visium slide would take more than 17 days. Since we analyzed many datasets in this paper, it is not practical to use XFuse to enhance gene expression resolution. In addition, XFuse can only enhance gene expression resolution with high accuracy for about 100 genes, which makes it less useful for cell location recovery. Due to these reasons, we enhanced gene expression resolution only using TESLA.

The use of TESLA to enhance gene expression resolution is only a temporary solution. The best ST reference data are those that have a single-cell resolution and have whole-transcriptome coverage. While such data are not easily accessible, we anticipate they will become widely available in the near future. To demonstrate that CeLEry can utilize non-Visium data as ST reference, we also analyzed several other datasets in this revision. These new analyses include

- 1) Mouse brain MERSCOPE data (revised Fig. 6);
- 2) Liver cancer MERSCOPE data (revised Fig. 7);
- 3) Breast cancer 10x Xenium data (revised Fig. 8).

In all these data analyses, CeLEry outperformed Tangram, spaOTsc, and novoSpaRc.

14. On page 13, the authors wrote, “the reconstructed 2D gene expression maps based on CeLEry’s recovered 2D locations are more similar to the ground truth than those of Tangram (Fig. 7C).” However, in Fig 7C, it seems that the outputs from CeLEry and Tangram are quite similar visually. It is hard to tell which one is more similar to the ground truth. Can authors provide a quantification of this similarity to the ground truth?

We thank the reviewer for this constructive comment. Following your suggestion, we calculated the SSIM (structural similarity index measure) metric for each gene. SSIM is a commonly used metric to assess the similarity between two images. In this analysis, we treated the original gene expression map as an image and the reconstructed 2D gene expression map as another image. A higher SSIM indicates the reconstructed 2D gene expression is more similar to the original gene expression map. As shown in **Supplementary Fig. 5** (also pasted below), CeLEry outperformed the other methods.

Supplementary Fig. 5: Boxplot of structural similarity index (SSIM). Each dot represents the SSIM between the predicted and truth gene map for a single gene.

REVIEWER COMMENTS

Reviewer #1 (Remarks to the Author):

The authors have made a great effort to make the manuscript much better than the previous version. They have validated CeLEry on much more datasets from different tissues and compared its performance with other similar spatial reconstruction methods. Besides, they have also revised the Method section with more details. Most of my concerns have been addressed. However, some lingering concerns remain, most significantly with respect to the comprehensiveness of validation. My detailed comments are listed below, hoping these are useful to the authors.

1. The authors only applied multiple training data in the spatial domain recovery task. However, I wonder if this strategy can also effectively improve the accuracy of the spatial location recovery task. To validate this, the DLPFC datasets are optimal to use as the spatial sequencing reference is generated from the same experiment, which could help mitigate some batch effects in the spatial data. Then the authors can compare the results of CeLEry and other methods from the following experiment scenarios (Take "slice 151673 training, slice 151676 testing" as an example, and vice versa):

- Training data: slice 151673; Testing data: slice 151676
- Training data: slice 151673, 151674, 151675; Testing data: slice 151676
- Training data: slice 151673 (Augmentation); Testing data: slice 151676

2. Since the DLPFC datasets include 12 tissue slices from 3 brain donors, the authors should describe clearly which slices they used. In addition, does Figure 3b-d show the average results of all 12 slices or the results of only a single slice?

3. I appreciate that the authors benchmark CeLEry and other spatial reconstruction methods on multiple spatial transcriptomics datasets of different tissues, however, I found that the training dataset and testing dataset in each experiment are from the same tissue slice. For example, when analyzing the mouse brain MERSCOPE data, the training dataset was from the left brain and the testing dataset was from the right brain; and when analyzing the liver cancer MERSCOPE data, the data was randomly split into two parts and one for training and the remaining for testing. In my opinion, I don't think just splitting one slice data into the training and testing dataset is appropriate enough because tissue morphology and transcriptomics can be variable among biological replicates. Besides, in practice, the scRNA-seq data and spatial reference would be more likely to come from the different tissue slices. Therefore, I encourage the authors to employ different replicates of tissues to verify each other, i.e., training the model on a given ST data and testing over a separate ST data.

4. As a reference-based method, the analysis results of CeLEry may be affected by the spatial reference used. Is CeLEry conservative in recovering the spatial domain as well as the spatial location of scRNA-seq data when using spatial references generated from different experiments or labs? Please discuss about it in detail.

5. I'm very excited that there has been a significant improvement in spatial reconstruction performance of all methods, not just CeLEry, after applying TESLA (0.361 to 0.986 for CeLEry, and 0.187 to 0.975 for spaOTsc, etc.). The correlation of pairwise distances between cells in the true coordinates and predicted coordinates is above 0.9 is very surprising since predicting the absolute spatial locations of cells is a difficult task. I'm interested in this gene-enhancing step but fail to find any detailed descriptions in the Method section. In addition, I would encourage the authors to apply this gene-enhancing step on another dataset from different tissues such as tumors, and thus it will be possible to comprehend if the significant improvement (over 0.9) in space reconstruction after enhancing gene expression presented in the article is universal, or if instead it's only applicable to some specific data?

6. Would it be possible for the authors to validate the spatial reconstruction results through cell-cell communications (CCC)? If there is a known cell-cell communication that is spatially adjacent, then its ligand and receptor expression should be colocalized in the reconstructed space.

7. Figure 3a, Senario should be Scenario.

8. Figure 5c, the results of novoSpaRc also seem to be consistent with the ground truth. Are there any applicable metrics to compare the recovered gene expression map based on predicted locations by different methods?

Reviewer #2 (Remarks to the Author):

In Question 2, the authors decided to still use the same mean and standard deviation obtained from one slide to generate multiple slides for training the model. It is hard to be convinced that this is realistic. As shown in Supplementary Fig. 3 the three example genes, the experimentally generated biological replicates (the first rows) have obviously larger variability than the data augmentation results (second rows). The augmented data looks almost identical to the eyes. Since the training samples are too alike (generated using the same mean and sd from one slide), it is hard to believe that the model can be applied generally to scRNA-seq data — after all, the scRNA-seq data that the method is designed for inferring spatial locations was definitely sequenced from different donors and also in a different data modality. While the difference in data modality could be challenging to deal with, the difference in samples should not be ignored. Otherwise, the model seems to be not ready yet. It is suggested that the authors take a further step on this (either incorporate the difference in the model or justify the current procedure in a better way).

In Question 4, the mean of the gene expression has been scaled before being plotted (negative values exist). It will be helpful if it is marked in the figure caption so that users do not get confused (for both the mean and sd).

In Questions 7, 10, and 11, all those data examples have been deleted in the revised version unfortunately, and the authors added three new data examples.

For the new MERSCOPE mouse brain data, in Figure 6a, is the performance of spaOTsc better than that of CeLEry? For example, the cluster in (3000, 4800) in spaOTsc has lower entropy than that in CeLEry. In Figure 6b, the top ones (highlighted spots) are actually better in novoSpaRc (in around (2500, 2000)) than that in CeLEry (in around (2800, 2800)). A similar issue happens in the liver cancer data (Figure 7b) and breast cancer data (Figure 8b). Please justify the performance conclusion.

In Question 13, the authors replied, "it is not practical to use XFuse to enhance gene expression resolution. In addition, XFuse can only enhance gene expression resolution with high accuracy for about 100 genes, which makes it less useful for cell location recovery." However, in their newly added example, MERSCOPE mouse brain data, only around 100 expressed genes were kept for the analysis. Can the authors further justify why XFuse cannot be applied here? It is important that the intermediate step does not introduce confounding effects or bias in the evaluation. Otherwise, authors might want to note it in the discussion as a limitation.

POINT-TO-POINT RESPONSE TO REVIEWERS' COMMENTS

We would like to express our sincere gratitude to the reviewers for your valuable comments, which have greatly contributed to the improvement of our manuscript. Based on your feedback, we have made substantial revisions to our work, and have highlighted the revised and new sections in red for ease of reviewing. Below, we have provided a point-by-point response to the reviewers' comments. The original reviewers' comments are in **bold italics** and our responses are in normal font colored in blue.

The **major changes** that we made in this revision include the following:

1. Added three examples to evaluate CeLEry's performance when training and testing data are from different samples
 - MERSCOPE mouse brain data generated by Vizgen (revised **Fig. 6**);
 - MERFISH mouse brain data generated by Xiaowei Zhuang's lab (revised **Fig. 7**);
 - 10x Xenium breast cancer data (revised **Fig. 10**).

In these examples, we demonstrate that CeLEry is scalable to large datasets and its performance is improved when single-cell resolution ST data are used as spatial reference and when the number of cells in the training set is increased. We also showed that spaOTsc and novoSpaRc both have scalability issues and cannot analyze large datasets.

2. Added applications to demonstrate that CeLEry is able to predict 2D locations for scRNA-seq cells (revised **Fig. 8**).
3. Deleted the results reported in the original **Fig. 5e** because that analysis used 10x Visium data as spatial reference. Since we have added applications using MERSCOPE, MERFISH, and 10x Xenium data as the spatial references, we decided to delete this example.

Reviewer #1 (Remarks to the Author):

The authors have made a great effort to make the manuscript much better than the previous version. They have validated CeLEry on much more datasets from different tissues and compared its performance with other similar spatial reconstruction methods. Besides, they have also revised the Method section with more details. Most of my concerns have been addressed. However, some lingering concerns remain, most significantly with respect to the comprehensiveness of validation. My detailed comments are listed below, hoping these are useful to the authors.

Thanks for your encouraging comments and constructive critiques, which have helped us improve the quality of the work. Below, we address your comments in detail.

Major:

1. The authors only applied multiple training data in the spatial domain recovery task. However, I wonder if this strategy can also effectively improve the accuracy of the spatial location recovery task. To validate this, the DLPFC datasets are optimal to use as the spatial sequencing reference is generated from the same experiment, which could help mitigate some batch effects in the spatial data. Then the authors can compare the results of CeLEry and other methods from the following experiment scenarios (Take "slice 151673 training, slice 151676 testing" as an example, and vice versa):

- **Training data: slice 151673; Testing data: slice 151676**
- **Training data: slice 151673, 151674, 151675; Testing data: slice 151676**
- **Training data: slice 151673 (Augmentation); Testing data: slice 151676**

Thank you for this constructive suggestion. We completely agree that it would also be important to evaluate the multiple training data strategy for the spatial location recovery task. We used the DLPFC data for spatial domain recovery because this dataset has cortical layer annotation. However, this dataset is not ideal for the spatial location recovery task because it doesn't have single-cell resolution. To evaluate the performance of CeLEry

similar to the scenarios that you suggested, we conducted three different sets of analyses, all based on single-cell resolution ST data.

First, we analyzed the MERSCOPE data generated from mouse brain. This dataset includes 3 replicates (sample IDs: S1R1, S1R2, and S1R3), which allowed us to conduct analyses using the designs suggested by you. Below is the description of our results for these analyses (lines 298-341 in the revised manuscript):

For this dataset, we designed three scenarios as summarized in Fig. 6a. Scenario 1 involved one replicate (sample ID: S1R1) containing 78,329 cells and 649 genes. After data quality control, we kept 18,342 cells with more than 100 expressed genes and 500 total UMI counts in analysis. We were stringent in filtering because MERSCOPE does not cover the entire transcriptome and many of its measured genes were not expressed or were only expressed in a few cells. In this analysis, we split the data into two halves (left and right brain), using cells from the left brain for training and cells from the rest brain for testing, and vice versa. This allowed us to recover the cell location information for the entire brain. Furthermore, since this MERSCOPE dataset includes two other replicates for the same slice (S1R2 with 88,884 cells and S1R3 with 84,636 cells), it provided an opportunity to evaluate the performance of CeLEry in more complicated situations. Therefore, in Scenarios 2 and 3, we ensured that the training set and testing set were from different replicates. In Scenario 2, we used cells in the right brain of replicate S1R2 for training and tested on cells in S1R1. Since mouse brains are symmetric, it is rational to use one half of the brain to train the model and obtain predictions for both halves of the brain in the testing set. In Scenario 3, we merged the right brains of replicates S1R2 and S1R3, as their locations matched well, to enlarge the sample size of the training set and introduce more variable samples, and used the same testing set as Scenarios 2 (Fig. 6b).

We first examined whether CeLEry could retain the pairwise spatial distance between every two cells. We calculated the Pearson correlation between the true pairwise distance and the predicted pairwise distance for all cell pairs in this dataset. As shown in Fig. 6c,d, in Scenario 1, CeLEry achieved a correlation of 0.72, outperforming Tangram, spaOTsc, and novoSpaRc, which had correlations of only 0.38, 0.67, and 0.56, respectively. In Scenario 2, where we applied the trained model to a different replicate, CeLEry still demonstrated robust performance with a correlation of 0.69. While spaOTsc also maintained stable prediction power with a correlation of 0.64, the correlations of Tangram and novoSpaRc dropped to 0.24 and 0.40, respectively, demonstrating that they are sensitive to variability between different replicates. When multiple training replicates were used (Scenario 3), CeLEry's performance improved even further, achieving a correlation of 0.74, which is even higher than when the training set and testing set were from the same replicate (Scenario 1).

To better understand the accuracy of cell location recovery in different regions, we calculated the Euclidean distance between the true and predicted absolute locations for each cell in the test data and plotted the distribution as boxplots. However, in Scenarios 2 and 3, the predicted locations were in the space of the training replicate(s) and had a different orientation from the ground truth in Scenario 1. To make them comparable, we rotated the ground truth in Scenario 1 to roughly match their original orientation. Due to imperfect matching, it was not fair to directly compare Scenario 1 with Scenarios 2 and 3. However, we can still compare the performance of different methods within a single scenario or between Scenarios 2 and 3, which taking into account the effect of imperfect matching. The results presented in Fig 6e,f show that CeLEry outperformed Tangram and novoSpaRc, and was on par with spaOTsc in all scenarios. However, both CeLEry and spaOTsc struggled to map cells from similar regions. In Scenario 3, the Euclidean distances between predicted and true locations were generally lower than in Scenario 2, suggesting that increasing the training sample size improved prediction performance. Fig. 6g and Supplementary Fig. 8 show randomly selected genes, whose spatial gene expression patterns were well recovered by CeLEry and spaOTsc. These analyses indicate that the location-recovered single-cell data from CeLEry can recapitulate the original spatial expression patterns of the cells.

Figure 6 is shown below:

Fig. 6. 2D location recovery for single cells in MERSCOPE mouse brain data. **a**, Three scenarios with varying degrees of complexity were considered for benchmark evaluations. **b**, Overlay of the three replicates based on their spatial coordinates. **c**, Barplot of Pearson correlation between true and predicted pairwise distances for all cell pairs. **d**, Scatter density plot comparing true and predicted pairwise distances for all pairs in Scenario 3. Color in the plot indicates density

of cell pairs. **e**, Boxplot of Euclidean distances between true and predicted locations for all cells in the test data. **f**, Visualization of Euclidean distances between true and predicted locations for all cells in the test data for Scenario 3. **g**, Recovered gene expression map of a randomly selected gene *ADRB1*, based on the predicted locations by CeLEry in Scenario 3, with color indicating relative gene expression.

Second, we analyzed a recently released MERFISH dataset generated from mouse brain by Zhang *et al.* (<https://www.biorxiv.org/content/10.1101/2023.03.06.531348v1>). We selected 3 tissue slices (sample IDs: slice 4, slice 5, and slice 7) from this study and conducted analyses using the designs suggested by you. Below is the description of our results from these analyses (lines 343-380 in the revised manuscript).

Motivated by the promising performance of CeLEry when using single-cell resolution ST reference in the mouse brain, we explored another single-cell resolution ST reference dataset that has more cells and genes. Specifically, we analyzed the Mouse3_sagittal dataset generated by Zhang et al. using MERFISH. One advantage of this dataset is that it includes 1147 genes, which is more than the previous MERSCOPE dataset, providing more genes for location recovery. Additionally, it includes 25 coronal slices, allowing us to examine the scenario where the training and testing sets are from different tissue slices. We selected three slices (Slice ID: sa1_slice4, sa1_slice5, sa1_slice7) with similar location regions and orientation to evaluate the performance of CeLEry. The data quality control criteria were the same as the study of mouse brain MERSCOPE data.

In Fig 7a, we considered two scenarios for evaluating the performance of CeLEry on the Mouse3_sagittal dataset. In Scenario 1, we used cells in slice 5 as the training set and cells in slice 7 as the testing set. In Scenario 2, we combined cells from slice 5 and slice 4 to form the training set and evaluated the performance of CeLEry on the same test data as Scenario 1. However, because slice 4 and slice 5 had some discrepancies in their location regions, we shifted the locations of slice 4 to approximately align them with slice 5, as shown in Fig. 7b. Due to the large number of cells in the dataset, spaOTsc and novoSpaRc were not able to handle Scenario 2. To evaluate the prediction accuracy, we compared the true pairwise distances with the predicted pairwise distances for all cell pairs and calculated the Pearson correlation. In Scenario 1, CeLEry reached a correlation of 0.89, outperforming Tangram (0.68), spaOTsc (0.81), and novoSpaRc (0.71) (Fig. 7c,d). Expanding the training set in Scenario 2 led to improved accuracy of CeLEry, with a correlation of 0.90, while Tangram's correlation fell to 0.63. To investigate the spatial variability in prediction accuracy across spatial locations, we calculated the Euclidean distance between the true and predicted locations for each cell of the test data. As shown in Fig. 7e, CeLEry outperformed Tangram, spaOTsc, and novoSpaRc in both scenarios, with generally lower discrepancies between true and predicted locations. Furthermore, Fig. 7f revealed that CeLEry maintained a consistent level of prediction accuracy across all locations, while Tangram exhibited spatial variability and performed poorly in some regions.

We conducted further validation of our spatial reconstruction results by analyzing cell-cell communication patterns. To retrieve known ligand-receptor pairs, we utilized CellChatDB, an existing database for storing ligand-receptor pairs. We selected several pairs whose gene expression showed spatial patterns and compared their expression patterns for ligand and receptor under true locations and locations predicted by CeLEry. As depicted in Fig. 7g and Supplementary Fig. 9c, CeLEry was able to recover the expression patterns of the ligand and receptor pairs, with their expression exhibiting colocalized patterns in certain regions of the brain, suggesting potential cell-cell communications in those regions.

Figure 7 is shown below:

Fig. 7. 2D location recovery for single cells in MERFISH mouse brain data. **a**, Two scenarios with varying degrees of complexity were considered for benchmark evaluations. **b**, Overlay of the two training slices based on their original spatial coordinates (left) and coordinates after manual location matching (right). **c**, Barplot of Pearson correlation between true and predicted pairwise distances for all cell pairs. SpaOTsc and novoSpaRc were not able to handle the large number of cells in Scenario 2. **d**, Scatter density plot comparing true and predicted pairwise distances for all cell pairs in Scenario 2. Color in the plot indicates density of cell pairs. **e**, Boxplot of Euclidean distances between true and predicted locations for all cells in the test data. SpaOTsc and novoSpaRc were not able to handle the large number of cells in Scenario 2. **f**, Visualization of Euclidean distance between the true and predicted locations for each cell in the test data for Scenario 2. **g**, True and recovered gene expression maps of two ligand-receptor pairs, based on the predicted locations by CeLEry, with color indicating relative gene expression for Scenario 2.

Third, we analyzed the recently released 10x Xenium breast cancer data, which include 2 tissue slices. We conducted evaluations following the designs suggested by you. Below is the description of our results from these analyses (lines 433-507 in the revised manuscript).

Building on the success of CeLEry in analyzing the MERSCOPE liver cancer data, we decided to investigate its applicability in another cancer dataset. Specifically, we analyzed a HER2-positive, ESR1-positive, PGR-negative breast tumor dataset obtained from the recently released 10x Xenium platform²⁹. This dataset contains information from two consecutively cut tissue sections from a breast cancer patient. Since the Xenium breast cancer data include two tissue sections, this allowed us to evaluate the performance of CeLEry when the training and test data are from different tissue sections. In our analysis, we utilized the slightly larger tissue section, Replicate 1, consisting of 167,782 cells for training the models and the smaller section, Replicate 2, consisting of 118,708 cells for validation. Both of the Xenium In Situ breast cancer tissue replicates contain data on a panel of 313 matching genes. Since this dataset is noisy and has a limited number of genes, we filtered out cells in Replicate 2 that ranked in the bottom 75% for total UMI counts and the bottom 25% for the number of expressed genes for the 2D coordinate predictions. This resulted in a total of 29,770 cells in the validation set for our analysis. Because the competing methods were unable to handle the computational load of this task, we had to perform subsampling of the cells prior to running these methods. We ensured to maintain the same training to test size ratio that was used for CeLEry and subsampled 50,000 cells for both the novoSpaRc and Tangram analyses and 40,000 cells for the SpaOTsc analyses based on the computational capacity of the methods.

*The Xenium In Situ Replicate 1 training dataset is at single-cell resolution. However, in practice, users may not have access to a single-cell resolution dataset for model training for their specific question of interest. Therefore, we considered three schemes to simulate different training scenarios that a user may face when attempting to recover the 2D architecture of their data (**Fig. 10a**). The first scenario is the baseline where we used the original Xenium single-cell resolution dataset for model training. Next, for Scenario 2, we generated an artificial Visium dataset from the Xenium single-cell Replicate 1 dataset to represent a lower-resolution spot-level gene expression dataset containing 4,215 spots, similar to what users may have access to and use for model training in practice. For Scenario 3, we enhanced the gene expression resolution of the artificial Visium training dataset from Scenario 2 using TESLA²³ in order to assess improvements in prediction performance from increasing the gene expression resolution when a histology image is available.*

*When assessing the performance of the methods across the three scenarios, we first analyzed the relationship between the true and predicted pairwise distances among each cell. The results, depicted in **Fig. 10b** and **Supplementary Fig. 11a**, show that CeLEry achieved significantly higher correlations of 0.74, 0.46, and 0.51 between the true and predicted pairwise distances across all three scenarios respectively compared to Tangram (0.33, 0.2, 0.25), spaOTsc (0.41, 0.35, 0.36), and novoSpaRc (0.43, 0.39, 0.42). We also see the relationship between the true and predicted pairwise distances from CeLEry much more closely follow a diagonal 45-degree line in all three scenarios compared to the other competing methods as shown*

in **Fig. 10c**, **Supplementary Fig. 12a,c**. Here we see using the original single-cell resolution Xenium data for model training results in the highest pairwise correlation for all methods (Scenario 1). The results show an improvement in the reconstruction of pairwise distances when using the TESLA enhanced gene-expression across all methods (Scenario 3) compared to the artificial Visium spot-level resolution data (Scenario 2).

To further explore the spatial heterogeneity of the prediction accuracy, we calculated the Euclidean distance between the true and predicted locations of each cell in the test data for each of the proposed scenarios. The results, shown in **Fig. 10d**, indicate that coordinates recovered by CeLEry have an overall lower Euclidean distance from the true coordinates compared to the coordinates recovered by Tangram, spaOTsc, and novoSpaRc. The median Euclidean distance between the true and predicted coordinates from CeLEry were 1,645, 2,659, and 1,995 across the three scenarios respectively. The median Euclidean distance between the true and predicted coordinates were considerably higher for Tangram (2,632, 3,458, 3,133), spaOTsc (2,651, 3,302, 2,948) and novoSpaRc (2,564, 3,349, 2,863) for all three scenarios. The results indicate that using the single-cell Xenium In Situ data for model training results in predictions with the lowest Euclidean distance from the truth (**Fig. 10d**). When comparing Scenarios 2 and 3, the results show that performing gene resolution enhancement on the Visium spot-level data leads to improvement in overall prediction with predicted locations being closer to the true locations across all methods. We note that although CeLEry does tend to have a larger range in the Euclidean distances compared to the other methods, we have found Tangram, spaOTsc, and novoSpaRc are all prone to predicting cells locations very close to the tissue center resulting in smaller bounds in the estimates. As shown in **Fig. 10e**, **Supplementary Fig. 12b,d**, CeLEry consistently performed well across different locations in the tissue, whereas Tangram, spaOTsc, and novoSpaRc, produced larger errors when predicting cells located far from the tissue center for all scenarios investigated. Furthermore, **Fig. 10f** and **Supplementary Fig. 13** depict the recovered spatial gene expression patterns for selected genes showing various spatial patterns from the Xenium panel including breast cancer genes ERBB2 and ESR1 for biological interest since the tissue section is classified as HER2 and ER positive. Additionally, the tissue structure shown in **Fig. 10a** demonstrates a higher degree of cell heterogeneity on the x-axis than the y-axis. Therefore, we see that it was generally easier to predict the x-axis than the y-axis, as the correlation of truth and predicted coordinates on the x-axis was higher than that of the y-axis (**Supplementary Fig. 11b**) for this type of tissue architecture. Overall, we show that CeLEry is able to successfully reconstruct the tissue architecture when training and testing from two different tissue sections. The results, as shown in **Fig. 10a-c**, indicate that using single-cell resolution data (Scenario 1), when available, for model training outperforms using Visium spot-level or spot-level enhanced data, as expected. However, when single-cell resolution data is not available for training, we have found that utilizing a gene resolution enhancing algorithm, such as TESLA²³, on the training data prior to predicting the 2D location coordinates results in better tissue reconstruction compared to training on Visium spot-level data for all methods evaluated.

Figure 10 is shown below,

Fig. 10. 2D location recovery for single cells in 10X Xenium breast cancer data. **a**, Visualization of Replicate 1 and the three training scenarios investigated with varying spatial resolutions including Xenium single-cell, artificial Visium spot-level, and enhanced spot-level along with visualization of the single-cell Xenium Replicate 2 test dataset. **b**, Side-by-side bar graph of Pearson correlation between true and predicted pairwise distances across all cell pairs for each method and scenario evaluated. **c**, Scatter density plots comparing true and predicted pairwise distances for all cell pairs in Scenario 1. Color in the plot indicates the density of cell pairs. **d**, Side-by-side boxplots of Euclidean distances between true and predicted locations for all cells in the test data for each method and scenario evaluated. **e**, Visualization of Euclidean distances between true and predicted locations for all cells in the test data for Scenario 1. **f**, Recovered gene expression map for Scenario 1 for a randomly selected gene, *GATA3*, with color indicating relative gene expression.

2. Since the DLPCF datasets include 12 tissue slices from 3 brain donors, the authors should describe clearly which slices they used. In addition, does Figure 3b-d show the average results of all 12 slices or the results of only a single slice?

We apologize for the lack of clarity. Following your suggestion, we have provided details about the sample IDs in **Fig. 3a**, which is also pasted below.

Fig. 3b-d showed the results for a single test slice, and the results were obtained from the 4 different scenarios described in **Fig. 3a**. For example, in **Fig. 3d** (also pasted below),

Scenarios 1 and 3 is the performance for test slice 151676, whereas Scenarios 2 and 4 were results for test slice 151507. We have labeled the slice ID for the training and testing data in **Fig. 3a** to make the message clear. We hope these revisions make it easier to interpret the results.

3. I appreciate that the authors benchmark CeLEry and other spatial reconstruction methods on multiple spatial transcriptomics datasets of different tissues, however, I found that the training dataset and testing dataset in each experiment are from the same tissue slice. For example, when analyzing the mouse brain MERSCOPE data, the training dataset was from the left brain and the testing dataset was from the right brain; and when analyzing the liver cancer MERSCOPE data, the data was randomly split into two parts and one for training and the remaining for testing. In my opinion, I don't think just splitting one slice data into the training and testing dataset is appropriate enough because tissue morphology and transcriptomics can be variable among biological replicates. Besides, in practice, the scRNA-seq data and spatial reference would be more likely to come from the different tissue slices. Therefore, I encourage the authors to employ different replicates of tissues to verify each other, i.e., training the model on a given ST data and testing over a separate ST data.

We completely agree with the reviewer that a more realistic evaluation is to use different tissue slices for the training and test data. In this revision, we have conducted such evaluations. Please see our response to your comment #1, where we included the results from analyses of the MERSCOPE and MERFISH mouse brain data, and 10x Xenium breast cancer data.

In addition to the above evaluations, we also considered a real application where the test data are from a scRNA-seq study. In this analysis, we used CeLEry to predict the 2D coordinates of cells in a mouse brain scRNA-seq dataset generated by the Allen Brain Atlas and assessed its reliability and robustness with different ST references. Below is the description of the results from these analyses (lines 382-401 in the revised manuscript).

*To demonstrate the practical utility of CeLEry in predicting 2D cell locations, we applied it to a scRNA-seq dataset obtained from mouse brain. For illustration purposes, we randomly sampled 500 cells from cortical layers, including L2 IT ENT1, L2/3 IT CTX, L4/5 IT CTX, L5 IT CTX, L6 CT CTX, L6b CT ENT, resulting in a total of 3000 cells. Our objective was to assess the reliability of CeLEry in predicting the 2D locations of these cells. To evaluate the robustness of the model with different ST references, we conducted two studies: one using the MERSCOPE mouse brain data generated by Vizgen as the reference and the other using the MERFISH mouse brain data generated by Zhang et al. as the reference. We trained the models on these reference data to learn the relationship between gene expression and 2D coordinates and then utilized the trained model to predict the locations of the sampled cells from the scRNA-seq data. As shown in **Fig. 8**, the cell locations predicted by CeLEry revealed a clear cortical layer structure that aligned with the known cortical layer information of these cells, irrespective of the ST reference utilized. In contrast, the locations predicted by Tangram, spaOTsc, and novoSpaRc did not correspond well with the expected results, with many cells predicted to wrong locations.*

Below is **Fig. 8**,

Fig. 8. Application to recover 2D locations for scRNA-seq data in mouse brain with different spatial references. a, 2D locations for scRNA-seq cells predicted by CeLEry, Tangram, spaOTsc, and novoSpaRc, using Vizgen’s MERSCOPE mouse brain data as the spatial reference. **b,** 2D locations for scRNA-seq cells predicted by CeLEry, Tangram, spaOTsc, and novoSpaRc, using MERFISH mouse brain data generated by Zhang et. al. as the spatial reference.

4. As a reference-based method, the analysis results of CeLEry may be affected by the spatial reference used. Is CeLEry conservative in recovering the spatial domain as well as the spatial location of scRNA-seq data when using spatial references generated from different experiments or labs? Please discuss about it in detail.

Thanks for this thoughtful comment. Please see our response to your comments #1 and #3, where we have added applications in which the training and testing data are from different samples or are generated by different technologies. In all applications, CeLEry demonstrated promising performance in recovering the true locations of cells.

5. I’m very excited that there has been a significant improvement in spatial reconstruction performance of all methods, not just CeLEry, after applying TESLA (0.361 to 0.986 for CeLEry, and 0.187 to 0.975 for spaOTsc, etc.). The correlation of pairwise distances between cells in the true coordinates and predicted coordinates is above 0.9 is very surprising since predicting the absolute spatial locations of cells is a difficult task. I’m interested in this gene-enhancing step but fail to find any detailed descriptions in the Method section. In addition, I would encourage the authors to apply this gene-enhancing step on another dataset from different tissues such as tumors, and thus it will be possible to comprehend if the significant improvement (over 0.9) in space reconstruction after enhancing gene expression presented in the article is universal, or if instead it’s only applicable to some specific data?

Thanks for this insightful comment. There are several reasons that led to the high correlations after applying TESLA.

1. TESLA increased the resolution from 872 spots to 5824 superpixels (each superpixel is 50x50 pixels) in the mouse brain dataset analyzed in Fig. 5. Since there are >70% tissue gaps in the Visium data that are not covered by spots, after doing TESLA gene expression resolution enhancement, we not only enhanced resolution within directly measured spots, but also filled in tissue gaps with super-resolution gene expression. As such, the TESLA enhanced data provide a more refined spatial reference, which allows all methods to better learn the relationships between gene expression and spatial locations.

- The tissue area that we considered is the cerebellum region in mouse brain. As shown in **Fig. 5c**, this region has distinct spatial structure, which made it easier to map the scRNA-seq cells to the region.

Due to the above reasons, the correlations became high for all methods. However, we would like to note that such high correlations are not universally true. To demonstrate this, we performed an experiment based on the recently released 10x Xenium breast cancer dataset. In this analysis, we generated an artificial Visium dataset from the Xenium single-cell *Replicate 1* dataset to represent a lower-resolution spot-level gene expression dataset containing 4,215 spots, similar to what users may have access to and use for model training in practice. We then enhanced the gene expression resolution of the artificial Visium training dataset using TESLA in order to assess improvements in prediction performance from increasing the gene expression resolution when a histology image is available. We found an improvement in the reconstruction of pairwise distances when using the TESLA enhanced gene expression across all methods compared to the artificial Visium spot-level resolution data. The results are shown below (also in Figure 10).

However, the degree of improvement is not as pronounced as the cerebellum region in mouse brain because the tissue structure in this breast cancer dataset is much more complex. Therefore, we expect to see improvement of all methods after gene expression resolution enhancement, but depending on the tissue type, the degree of improvement is generally modest.

6. Would it be possible for the authors to validate the spatial reconstruction results through cell-cell communications (CCC)? If there is a known cell-cell communication that is spatially adjacent, then its ligand and receptor expression should be colocalized in the reconstructed space.

Thanks for this very constructive comment. Following your suggested, we examined the expression patterns of known ligand-receptor pairs, and below is the description of our results for this analysis.

*We conducted further validation of our spatial reconstruction results by analyzing cell-cell communication patterns. To retrieve known ligand-receptor pairs, we utilized CellChatDB, an existing database for storing ligand-receptor pairs. We selected several pairs whose gene expression showed spatial patterns and compared their expression patterns for ligand and receptor under true locations and locations predicted by CeLEry. As depicted in **Fig. 7g** and **Supplementary Fig. 9c**, CeLEry was able to recover the expression patterns of the ligand and receptor pairs, with their expression exhibiting colocalized patterns in certain regions of the brain, suggesting potential cell-cell communications in those regions.*

Fig. 7g is shown below,

7. Figure 3a, Senario should be Scenario.

Thank you for catching this typo. We have corrected it in the revised version. The new Fig. 3a is pasted below.

8. Figure 5c, the results of novoSpaRc also seem to be consistent with the ground truth. Are there any applicable metrics to compare the recovered gene expression map based on predicted locations by different methods?

Thank you for this insightful comment. In the previous revision, we calculated the SSIM (structural similarity index measure) metric for each gene. SSIM is a commonly used metric to assess the similarity between two images. In this analysis, we treated the original gene expression map as an image and the reconstructed 2D gene expression map as another image. A higher SSIM indicates the reconstructed 2D gene expression is more similar to the original gene expression map. As shown in **Supplementary Fig. 5** (also pasted below), CeLEry outperformed the other methods.

Supplementary Fig. 5: Boxplot of structural similarity index (SSIM). Each dot represents the SSIM between the predicted and truth gene map for a single gene.

Reviewer #2 (Remarks to the Author):

Here are other comments.

1. In Question 2, the authors decided to still use the same mean and standard deviation obtained from one slide to generate multiple slides for training the model. It is hard to be convinced that this is realistic. As shown in Supplementary Fig. 3 the three example genes, the experimentally generated biological replicates (the first rows) have obviously larger variability than the data augmentation results (second rows). The augmented data looks almost identical to the eyes. Since the training samples are too alike (generated using the same mean and sd from one slide), it is hard to believe that the model can be applied generally to scRNA-seq data — after all, the scRNA-seq data that the method is designed for inferring spatial locations was definitely sequenced from different donors and also in a different data modality. While the difference in data modality could be challenging to deal with, the difference in samples should not be ignored. Otherwise, the model seems to be not ready yet. It is suggested that the authors take a further step on this (either incorporate the difference in the model or justify the current procedure in a better way).

We apologize for the lack of clarity in the previous revision, which might have caused confusion. The data augmentation step generates ST replicates based on the learned embedding distribution, which is characterized by the mean and standard deviation of a normal distribution. Therefore, any ST replicated generated from this embedding distribution will be sampled from this normal distribution with the same mean and standard deviation. This is the standard data augmentation based on generative modeling in machine learning. Although we described data augmentation in the paper, this step is only an optional step in CeLEry, i.e., CeLEry can still be run without doing data augmentation. We added this data augmentation option because some of the ST data, e.g., a single tissue slice in Visium, may not have enough number of spots for the model to reliably learn the relationship between gene expression and spatial locations. As such, there is a possibility of overfitting because the model hasn't seen enough variabilities in the training set. Our purpose of doing data augmentation is to introduce extra variations in the training data so that the location prediction model is more robust to training and test data differences. As we showed in **Fig. 5d** (also pasted below),

when the degree of gene expression noise in the test data increases, the differences between the training and test data also increase, and the augmented CeLEry model outperformed the ordinary CeLEry model without data augmentation. This example demonstrates that the data augmentation step can help improve the robustness of the prediction model to noise in the test data.

Since the original version of our paper was submitted in July 2022, there have been more and more single-cell resolution ST data became available. These single-cell ST data have large numbers of cells, and thus the data augmentation step is no longer needed for these data. In the previous revision and this revision, we have added

several additional examples where the ST references were generated using MERSCOPE, MERFISH, and 10x Xenium platforms, all have single-cell resolution. In these examples, CeLery without data augmentation was used in data analyses. Therefore, even if the data augmentation step does not fully capture all of the variations in real biological samples, it won't affect the performance of CeLery as long as single-cell resolution ST data are used as references or the training Visium ST data have enough number of spots.

Nevertheless, we recognize that our data augmentation step has limitations and by no means it can replace the real biological samples. In the Discussion, we have pointed out the limitations of data augmentation and made it clear that it only serves as an optional step when analyzing ST training data that have limited numbers of spots or cells. Below is the corresponding text in the Discussion (lines 575-578 in the revised manuscript):

Our data augmentation procedure is optional and was designed for ST data that have a grid layout, e.g., 10x Visium. ST data with non-grid layout, e.g., SLIDE-seq, MERSCOPE, MERFISH, or 10x Xenium, would not benefit from this procedure. However, since the number of cells in MERSCOPE, MERFISH, and Xenium data is typically large, data augmentation is unnecessary.

2. In Question 4, the mean of the gene expression has been scaled before being plotted (negative values exist). It will be helpful if it is marked in the figure caption so that users do not get confused (for both the mean and sd).

Thank you for this comment. Following your suggestion, we have now added the mark in the figure caption.

3. In Questions 7, 10, and 11, all those data examples have been deleted in the revised version unfortunately, and the authors added three new data examples. For the new MERSCOPE mouse brain data, in Figure 6a, is the performance of spaOTsc better than that of CeLery? For example, the cluster in (3000, 4800) in spaOTsc has lower entropy than that in CeLery. In Figure 6b, the top ones (highlighted spots) are actually better in novoSpaRc (in around (2500, 2000)) than that in CeLery (in around (2800, 2800)). A similar issue happens in the liver cancer data (Figure 7b) and breast cancer data (Figure 8b). Please justify the performance conclusion.

We apologize for not providing a clear explanation regarding the removal of the data examples mentioned in your previous Questions 7, 10, and 11. We would like to clarify that these examples were originally obtained from 10x Visium data. However, we have since acquired many more examples using the single-cell resolution ST data generated from MERSCOPE, MERFISH, and 10x Xenium. We believe that these newer results are much stronger than the previous ones. As we explained in the manuscript, the ideal ST reference for CeLery is single-cell resolution ST data. Therefore, based on these reasons, we decided to delete unnecessary 10x Visium examples.

Also, thanks for your great questions about **Fig. 6a** and **Fig. 6b**. Below is the original **Fig. 6a** (now in **Supplementary Fig. 7a**),

It is true that “the cluster in (3000, 4800) in spaOTsc has lower entropy than that in CeLery.” However, visually appealing results do not necessarily mean that the cells are mapped to the correct location. This is because we

colored the cells according to their cluster assignment. To understand why visual examination based on Fig. 6a alone is not sufficient, let's consider an example. Suppose there is a cell that originates from location (3000, 4800) and that cell belongs to cluster 13 (grey). A correct mapping should map that cell to (3000, 4800). However, if we map another cell from cluster 13 to (3000, 4800) and color that cell with grey, we wouldn't be able to tell whether that is the correct cell or whether that is just another cell that comes from the grey cluster. Due to this reason, we further calculated the Euclidean distance between the true and predicted locations for each cell and visualized the Euclidean distances. We believe the Euclidean distances would better reflect which method predicts the location better. As shown below, where we color the cells according to the Euclidean distances,

we can see that CeLEry has comparable Euclidean distances with spaOTsc, which is reflected in the boxplots below

Regarding your question about the original Fig. 6b,

We are not sure why you think “the top ones (highlighted spots) are actually better in novoSpaRc (in around (2500, 2000)) than that in CeLEry (in around (2800, 2800)).” In the above plots, the color indicates cell pair density. novoSpaRc has a high density of cell pairs around (2500, 2000), but this indicates most of the cell pairs

that have the predicted distances differ from the true distances. This is because if a method could retain pairwise distances between cells, all cell pairs should fall along the 45-degree diagonal line; however, for novoSpaRc, we observed that the majority of the cell pairs fall below the 45-degree diagonal line, indicating its inability to retain pairwise distances between cells. This is further confirmed when calculating the Pearson correlation between the predicted and true pairwise distances, as shown below.

4. In Question 13, the authors replied, “it is not practical to use XFuse to enhance gene expression resolution. In addition, XFuse can only enhance gene expression resolution with high accuracy for about 100 genes, which makes it less useful for cell location recovery.” However, in their newly added example, MERSCOPE mouse brain data, only around 100 expressed genes were kept for the analysis. Can the authors further justify why XFuse cannot be applied here? It is important that the intermediate step does not introduce confounding effects or bias in the evaluation. Otherwise, authors might want to note it in the discussion as a limitation.

XFuse enhances gene expression resolution by utilizing high-resolution tissue information provided by histology images. Since the MERSCOPE data do not have the companion histology images, we cannot perform XFuse analysis on this dataset.

XFuse is a gene expression resolution enhancement tool. It is potentially useful when the spatial reference dataset does not have single-cell resolution, but for data generated from MERSCOPE, MERFISH, and Xenium, gene expression resolution enhancement is no longer needed because these platforms can generate spatial transcriptomics data with single-cell resolution.

REVIEWERS' COMMENTS

Reviewer #1 (Remarks to the Author):

The authors have addressed all my comments in the revision. I only have some minor remarks and suggestions.

1. In mouse brain MERSCOPE data analysis, if I understand correctly, the authors used the S1R1 Right as the training sample and the S1R1 Left as the validation sample in scenario 1. However, in scenario 2 and 3, they used S1R2 Right and S1R3 Right as the training sample respectively, but used the whole slice of S1R1 as the validation sample. In my opinion, I suggest that the authors unify the size of the training set and validation set in scenario 2 and 3, i.e., using the S1R2 Right (S1R3 Right) as the training sample and the S1R1 Left as the validation sample, or using the whole slice of S1R2 (S1R3) as the training sample and the whole slice of S1R1 as the validation sample.
2. I suggest again that the authors should take more concerned about choosing appropriate colors for the figures. For example, Figure 6g, 7g, and 9e, the colors represented 0 and 1 in the color bar are challenging to distinguish; Figure 8, the black border of cells covers the cell type colors, making it difficult to read.
3. Figure 6a and 10a, the cell type label is missing.
4. Figure 9, panels c and d should be interchanged.
5. Figure 10f, the color bar is missing.

Reviewer #2 (Remarks to the Author):

I appreciate the authors' time and effort in improving the quality of the manuscript. The clarifications on the pairwise distance are helpful. All my questions have been answered.

POINT-TO-POINT RESPONSE TO REVIEWERS' COMMENTS

We would like to express our sincere gratitude to the reviewers for your valuable comments, which have greatly contributed to the improvement of our manuscript. Based on your feedback, we have revised the manuscript, and highlighted the revised sections in red for ease of reviewing. Below, we provide a point-by-point response to the reviewers' comments. The original reviewers' comments are in **bold italics** and our responses are in normal font colored in blue.

Reviewer #1 (Remarks to the Author):

The authors have addressed all my comments in the revision. I only have some minor remarks and suggestions.

Thanks for carefully reviewing our manuscript. Below we address your remaining comments.

1. In mouse brain MERSCOPE data analysis, if I understand correctly, the authors used the S1R1 Right as the training sample and the S1R1 Left as the validation sample in scenario 1. However, in scenario 2 and 3, they used S1R2 Right and S1R3 Right as the training sample respectively, but used the whole slice of S1R1 as the validation sample. In my opinion, I suggest that the authors unify the size of the training set and validation set in scenario 2 and 3, i.e., using the S1R2 Right (S1R3 Right) as the training sample and the S1R1 Left as the validation sample, or using the whole slice of S1R2 (S1R3) as the training sample and the whole slice of S1R1 as the validation sample.

Thank you for this comment. For Scenario 1, indeed, we used the whole slice of S1R1 as the validation set, which was described in Lines 298-300, also shown below,

In this analysis, we split the data into two halves (left and right brain), using cells from the left brain for training and cells from the right brain for testing, and vice versa. This allowed us to recover the cell location information for the entire brain.

This procedure was indicated by the cross arrows in Figure 6a, which is also shown below,

With these explanations, we hope it is now clear to the reviewer that we used the same validation sample in all three scenarios and that the analyses are consistent across these scenarios.

2. I suggest again that the authors should take more concerned about choosing appropriate colors for the figures. For example, in Figure 6g, 7g, and 9e, the colors represented 0 and 1 in the color bar are challenging to distinguish; Figure 8, the black border of cells covers the cell type colors, making it difficult to read.

Thank you for this comment. Based on your suggestion, we have implemented new color schemes for Figures 6g, 7g, and 9e. Please find the updated figures below. We believe that the new color scheme offers improved contrast.

Figure 6g:

Figure 7g:

Figure 9e:

For Figure 8, we have made the cell size smaller and changed the black border of cells to a different color. The updated Figure 8 is shown below. We hope these changes make it easier to read.

3. Figure 6a and 10a, the cell type label is missing.

For Figures 6a and 10a, we don't have cell type labels as the original data that are publicly accessible didn't provide that information. The colors in Figures 6a and 10a represent clusters obtained from unsupervised clustering analysis conducted by us. We have explained this in the Figure legend, as shown below.

Figure 6. 2D location recovery for single cells in MERSCOPE mouse brain data. a Three scenarios with varying degrees of complexity were considered for benchmark evaluations. Color of each cell indicates cluster assignment obtained from unsupervised clustering.

Figure 10. 2D location recovery for single cells in 10X Xenium breast cancer data. a Visualization of Replicate 1 and the three training scenarios investigated with varying spatial resolutions including Xenium single-cell, artificial Visium spot-level, and enhanced spot-level along with visualization of the single-cell Xenium Replicate 2 test dataset. Color of each cell in the training sample and testing sample indicates cluster assignment obtained from unsupervised clustering.

4. Figure 9, panels c and d should be interchanged.

Thanks for catching this mistake. We have corrected Figure 9 following your suggestion. The updated Figure 9 is shown below.

5. Figure 10f, the color bar is missing.

Thank you! We have added the missing color bar in Figure 10f, which is shown below.

Reviewer #2 (Remarks to the Author):

I appreciate the authors' time and effort in improving the quality of the manuscript. The clarifications on the pairwise distance are helpful. All my questions have been answered.

Thank you!